# Pancreatic Ppy-expressing γ-cells display mixed phenotypic traits and the adaptive plasticity to engage insulin production

Marta Perez-Frances[1], Léon van Gurp [1], Maria Valentina Abate[1], Valentina Cigliola[1,4,5], Kenichiro Furuyama [1,2], Eva Bru-Tari [1], Daniel Oropeza[1], Taïna Carreaux[1], Yoshio Fujitani[3], Fabrizio Thorel[1] & Pedro L. Herrera [1✉]

The cellular identity of pancreatic polypeptide (Ppy)-expressing γ-cells, one of the rarest pancreatic islet cell-type, remains elusive. Within islets, glucagon and somatostatin, released respectively from α- and δ-cells, modulate the secretion of insulin by β-cells. Dysregulation of insulin production raises blood glucose levels, leading to diabetes onset. Here, we present the genetic signature of human and mouse γ-cells. Using different approaches, we identified a set of genes and pathways defining their functional identity. We found that the γ-cell population is heterogeneous, with subsets of cells producing another hormone in addition to Ppy. These bihormonal cells share identity markers typical of the other islet cell-types. In mice, *Ppy* gene inactivation or conditional γ-cell ablation did not alter glycemia nor body weight. Interestingly, upon β-cell injury induction, γ-cells exhibited gene expression changes and some of them engaged insulin production, like α- and δ-cells. In conclusion, we provide a comprehensive characterization of γ-cells and highlight their plasticity and therapeutic potential.

[1] Department of Genetic Medicine & Development, iGE3 and Centre facultaire du diabète, Faculty of Medicine, University of Geneva, Geneva, Switzerland. [2] Center for iPS Cell Research and Application (CiRA), Kyoto University, Kyoto, Japan. [3] Lab. of Developmental Biology & Metabolism, Institute for Molecular & Cellular Regulation, Gunma University, Maebashi, Gunma, Japan. [4] Present address: Department of Cell Biology, Duke University Medical Center, Durham, NC, USA. [5] Present address: Regeneration Next, Duke University, Durham, NC, USA. ✉email: pedro.herrera@unige.ch

Searching for precise schemes to classify the hundreds of cell-types that compose an organism, recent advances in genetic and cellular analyses have expanded our understanding of the cell identity concept. Striking heterogeneity has been reported in what used to be considered homogeneous cell populations[1,2], and new information continues to emerge from different cell systems. In particular, for endocrine cells, the production of specific hormones with highly specialized physiological functions form the basis of their identity. Various genetic pathways continue to be discovered that keep tight control of the production, processing and secretion of these endocrine hormones. Yet, surprisingly, there are reports of rare polyhormonal endocrine cells detected under different conditions. Bihormonal cells have been reported in the pituitary of fish, mice and humans[3–5]. Bihormonal cells have also been consistently reported during in vitro endocrine differentiation from human stem cells[6,7]. Polyhormonal cells are also present during mammalian pancreatic development (including humans) as well as in the adult endocrine pancreas, with increasing numbers in type 2 diabetic subjects[8,9]. Yet, these results suggest that bihormonal cells could be present in different endocrine organs and might have unexplored functions. In general, due to technical limitations, it has been hard to identify and characterize these types of cells. In our laboratory, using inducible cell lineage-tracing, we have reported that following near-total β-cell ablation, some 1–2% of mouse pancreatic islet cells engage insulin expression and become bihormonal[10,11].

In the adult murine pancreas, the islets of Langerhans contain different endocrine types of cells: glucagon-secreting α-cells, insulin-secreting β-cells, somatostatin-secreting δ-cells and pancreatic polypeptide (Ppy)-secreting γ-cells. A delicate and highly complex orchestration of intra-islet paracrine interactions is thought to be essential to achieve robust secretory dynamics and blood glucose homeostasis. α-cell signaling, via glucagon or GLP-1, can directly stimulate insulin secretion[12], while somatostatin directly inhibits insulin and glucagon secretion[13–15]. Providing balance, insulin indirectly inhibits glucagon release by promoting somatostatin secretion[14]. Given these and other observations, it is the consensus that hormone expression, and likely other key identity markers, is tightly controlled in each islet endocrine cell-type, with the near totality of cells producing only one of the main islet endocrine hormones[16,17]. Consequently, disruption of the pathways controlling endocrine cell-type identity and hormone production is thought to significantly impact islet cell function and might be one of the underlying causes of metabolic diseases like diabetes[18].

Compared to the other islet cell types, little is known about the function of γ-cells or the Ppy hormone, mainly due to their scarcity and a lack of appropriate tools to analyze them. Ppy is thought to be an anorexigenic factor, controlling gastric emptying, pancreatic and gallbladder secretion, and intestinal peristalsis[19–21]. Repeated administration of Ppy can reduce food intake and gastric emptying[22–25]. Yet, gain- and loss-of-function studies in mice have given confounding results[26–28], possibly due to compensatory mechanisms in the absence of Ppy signaling. Unfortunately, the mechanisms by which Ppy influences appetite, body weight and glucose homeostasis remain unclear. Yet, mounting evidence associates α- and δ-cells with the pathophysiology of diabetes, highlighting the role of islet non-β cells in blood glucose regulation[29–31].

Here, we describe three transgenic mouse lines to i) lineage-trace and ii) ablate Ppy-expressing γ-cells, and iii) inactivate the Ppy gene. While performing the in-depth characterization of these cells, we discovered that a significant proportion of them also expresses glucagon, somatostatin or insulin at the mRNA and protein levels. Interestingly, these bihormonal cells are also present in adult human islets. Incidentally, the silencing of Ppy gene and the ablation γ-cells provide insights into Ppy and γ-cell functions in controlling glucose homeostasis under basal conditions.

## Results

**Generation of a transgenic line to lineage-trace Ppy+ cells and inactivate the Ppy gene.** We developed a knock-in mouse line by replacing the coding sequence, in one allele, of the endogenous Ppy gene by the doxycycline (DOX)-dependent reverse transactivator (rtTA) coding region, using CRISPR technology (Fig. 1a, b). In these mice, rtTA expression is under the control of the endogenous Ppy gene regulatory elements. Triple Ppy-rtTA, TetO-Cre, Rosa26-STOP-YFP transgenic mice, termed Ppy-YFPi, were then generated to allow the inducible and irreversible expression of YFP upon DOX administration (Fig. 1c). This YFP tag allows to lineage-trace the Ppy-expressing cells (Fig. 1d, e;[32,33]). While no labeling was detected in absence of DOX (Supplementary Fig. 1; 1310 YFP+ cells scored; n = 3 mice; Source Data a), about 85% of Ppy-expressing cells were YFP-tagged in adult DOX-treated Ppy-YFPi mice either two weeks or ten months after DOX withdrawal (Fig. 1d, e; Source Data b). YFP-traced cells were Ppy-negative in Ppy-YFPi mice bearing the Ppy-rtTA transgene at homozygosity (Fig. 1f; Source Data b), confirming the inactivation of Ppy gene through the targeted insertion of rtTA and validating the specificity of the monoclonal Ppy antibodies (Supplementary Fig. 2). The distribution of YFP+ cells in DOX-treated Ppy-YFPi mice nicely recapitulates that of Ppy+ cells in wild type mice (WT) and humans, being more abundant in islets located in the head part of the pancreas[34,35] (Supplementary Fig. 3, Source Data c).

Thus, the Ppy-YFPi transgenic line allows for an efficient, inducible and irreversible labeling of Ppy-expressing γ-cells, and for inactivating the Ppy gene, at homozygosity. For this reason, in all subsequent experiments aimed at characterizing the Ppy-expressing γ-cells, the Ppy-rtTA allele was maintained at the heterozygous state, so as to preserve monoallelic wild-type Ppy expression in Ppy-YFPi mice.

**Adult γ-cells derive from embryonic Ppy-expressing cells.** We sought to determine whether adult Ppy-expressing cells are exclusively generated during embryonic pancreas development or also at postnatal stages. We thus performed a pulse-and-chase experiment aimed at labeling Ppy+ cells irreversibly in developing embryos, by dispensing DOX to pregnant females from E7.5 (2 days before the beginning of pancreas development) up to end of gestation (Fig. 2a). YFP-tracked progeny was analyzed in thirty-days-old (P30) Ppy-YFPi mice.

Labeling of embryonic Ppy+ cells was efficient in Ppy-YFPi embryos (80% of Ppy+ cells were YFP-labeled in near-term fetuses (E18.5); Supplementary Fig. 4, Source Data d). We confirmed that YFP labeling activity ceased rapidly after DOX withdrawal by evaluating the residual Cre mRNA expression as readout of DOX clearance in islets of pregnant Ppy-YFPi females. We found that Cre mRNA levels dropped to background levels by two days after DOX withdrawal (Supplementary Fig. 4, Source Data d), suggesting that DOX activity lasts for only about 24 h after DOX removal. Based on this observation, we defined the duration of DOX activity as the pulse period plus one additional day of clearance (Fig. 2a).

Next, we assessed the origin of the adult Ppy-expressing cells (one-month after birth, P30). Following DOX administration during pancreas development (E7.5 to E19.5), the proportion of YFP-tagged Ppy-expressing cells at P30 was equivalent to the labeling efficiency (~85%) observed in adult mice (Fig. 2b, Source Data e). This implies that adult γ-cells originate from Ppy+ cells appearing during pancreas development (i.e., prior to birth).

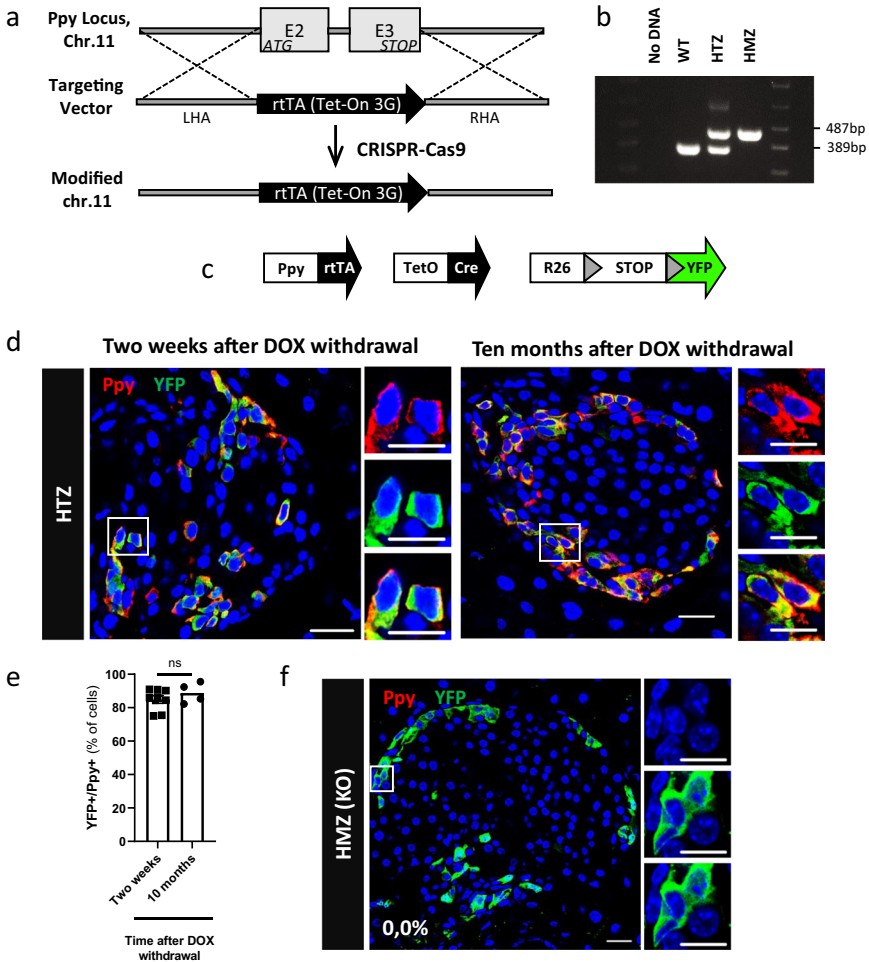

**Fig. 1 Generation of knock-in mice to lineage-trace γ-cells and inactivate the Ppy gene. a** CRISPR-Cas9 was used to replace the *Ppy* coding sequence (E2, E3: exons 2 & 3) on mouse chromosome 11 by the *rtTA* coding region ("targeting vector"). LHA left homology arm, RHA right homology arm. **b** PCR products of wild-type (WT), Ppy-rtTA heterozygous (HTZ) and Ppy-rtTA homozygous (HMZ) mice (WT band: 389 bp; KI band: 487 bp). "No DNA" is a negative control. Samples derive from the same experiment run at once in one gel. **c** Transgenes required for tracing the lineages of *Ppy*-expressing γ-cells. **d** Ppy+ cells are efficiently YFP-labeled two weeks or 10 months after DOX withdrawal in Ppy-YFPi mice. **e** Quantification of the YFP-labeled Ppy-expressing cells two weeks (84.9 ± 0.0, $n = 8$ mice, 5411 Ppy+YFP+ out of 6389 Ppy+ cell scored) or 10 months (88.8 ± 0.0%, $n = 4$ mice, 2852 Ppy +YFP+ out of 2889 Ppy+ cells scored) after DOX withdrawal. Data are shown as mean±s.e.m; two-tailed Mann–Whitney test ($P$ value 0.4606). **f** Ppy production is abrogated in Ppy-rtTA homozygous (HMZ, KO) mice: none of the YFP-labeled cells in DOX-treated animals (Ppy-YFPi, Ppy-rtTA HMZ) produces the peptidic hormone ($n = 3$ mice; 0 Ppy+ out of 1310 YFP+ cells scored). Ppy (red) and YFP (green). Scale bars: 20 μm (10 μm in insets). Region of the pancreas: Ventral. Source data are provided as Source Data file (Supplementary Table b).

Next, to evaluate Ppy+ cell emergence after P30, we extended the chase period until 9 months. Interestingly, we did not notice any decrease in the percentage of YFP-labeled adult cells if the chase period was extended up to 9 months after birth (Fig. 2b, Source Data e). Together, these observations indicate that adult γ-cells originate from Ppy+ cells appearing during pancreas development in utero, and that there is no evidence of postnatal Ppy+ cell neogenesis, at least nine months after birth.

We then evaluated whether the islet cells traced to an embryonic Ppy+ origin were producing other hormones instead of, or in addition to, Ppy. We found that some α- (glucagon+), δ- (somatostatin+) and β-cells (insulin+) were YFP-labeled in adult (one-month-old) Ppy-YFPi mice treated with DOX during gestation (Fig. 2c, d; Source Data e). Interestingly, some of these adult YFP+ cells also contained Ppy, and were thus bihormonal (Fig. 2c, d). We also observed embryonic YFP-traced cells co-expressing Ppy and other hormones during pancreas development (Supplementary Fig. 4). These results are consistent with previous observations showing that embryonic Ppy+ cells can

express other hormones and could contribute to other islet lineages[36–38].

**γ-cells are a heterogeneous population of mono- and bihormonal cells.** While Ppy+Gcg+ bihormonal cells were previously reported[39,40], co-expression of Ppy protein with either somatostatin or insulin in embryonic or adult islet cells was debated[38,41]. We thus studied the Ppy-expressing cells in adult Ppy-YFPi mice giving DOX to adult animals. Administration of DOX to one-month or one-year-old Ppy-YFPi mice resulted in the labeling of cells containing Ppy alone or in combination with either glucagon (~ 10% YFP+ cells were Gcg+), somatostatin (~ 4% YFP+ cells were Sst+) or insulin (~ 1% YFP+ cells were Ins+) (Fig. 3a, b; Supplementary Fig. 5; Source Data f). No significant statistical differences were found in the percentage of YFP-labeled hormone+ cells in one month or one-year-old mice (Fig. 3b; Source Data f). Ppy+Gcg+ and Ppy+Ins+ cells were the most and the least prevalent bihormonal cells, respectively (Fig. 3b; Source Data f). Co-expression of

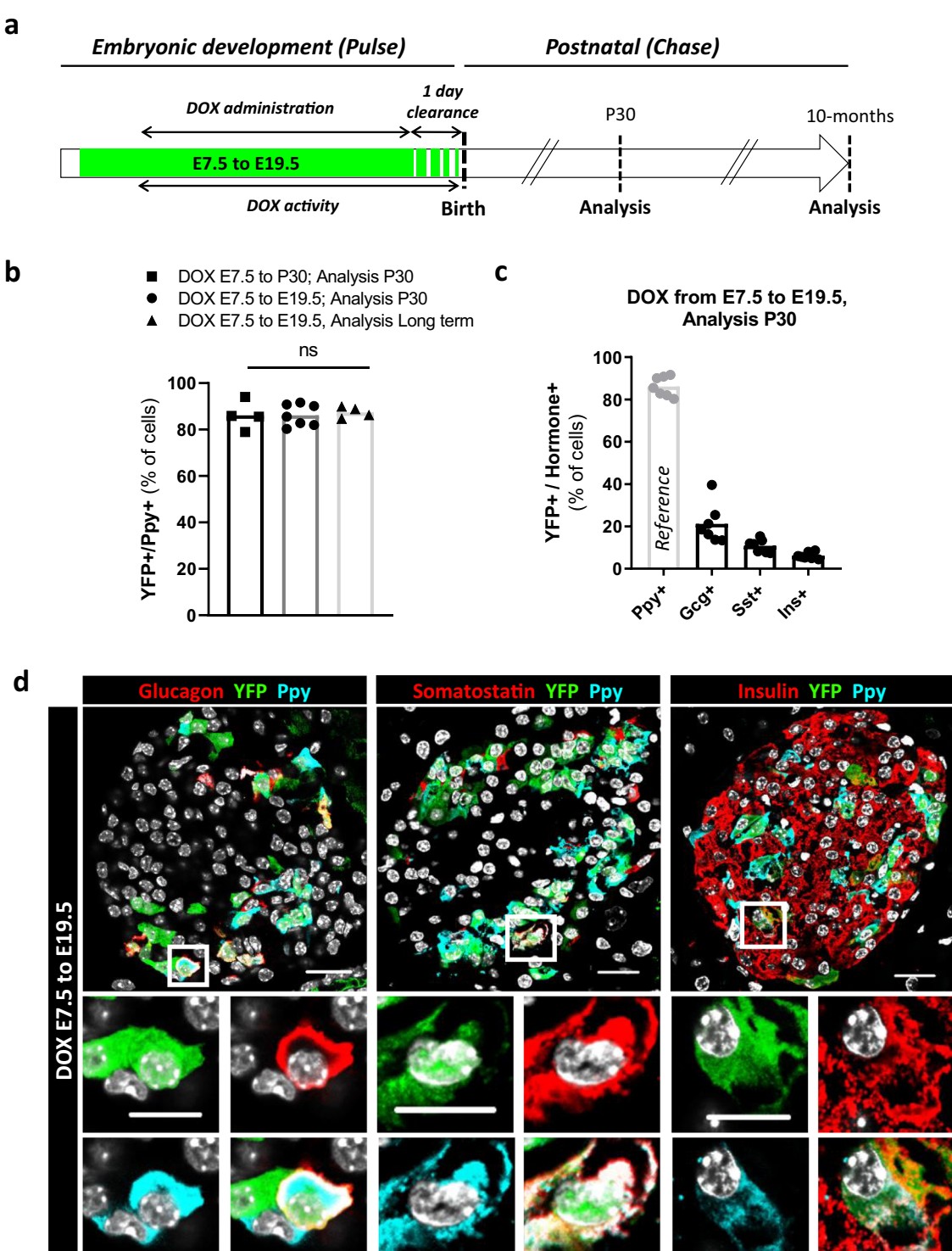

**Fig. 2 Adult γ-cells derive from embryonic Ppy-expressing cells. a** Embryonic Ppy-expressing cell labeling strategy. DOX doxycycline, P30 postnatal day 30. **b** The proportion of adult YFP-labeled Ppy-expressing cells is maintained in the different DOX administration periods. DOX E7.5 to P30, analysis P30 (black; $n = 4$ mice, 1842 Ppy+YFP+ out of 2131 Ppy+ cells scored); DOX E7.5 to E19.5, analysis P30 (dark gray, $n = 7$ mice, 3111 Ppy+YFP+ out of 3604 Ppy+ cells scored); DOX E7.5 to E19.5, analysis 9-month-old (light gray, $n = 4$ mice, 2241 Ppy+YFP+ out of 2565 Ppy+ cells scored). Data are shown as mean ± s.e.m; two-tailed Mann–Whitney test. P values: DOX E7.5 to P30, analysis P30 vs DOX E7.5 to E19.5, analysis P30 = 0.9273; DOX E7.5 to P30, analysis P30 vs DOX E7.5 to E19.5, analysis long term = 0.6857. **c** Percentage of adult hormone+ cells (Gcg+, Sst+ or Ins+) labeled with YFP when DOX was given during embryogenesis. 21.1 ± 0.0%, 10.8 ± 0.0% and 6.1 ± 0.0% of Gcg+, Sst+ and Ins+, respectively, were YFP-traced. The percentage of YFP-labeled Ppy+ cells is taken for reference from panel **b**. Data are shown as mean ± s.e.m; $n = 7$ mice; 810 YFP+Gcg+ out of 4196 Gcg+, 285 YFP+Sst+ out of 2669 Sst+ and 1306 YFP+Ins+ out of 20753 Ins+ cells scored. **d** Immunofluorescence of YFP (green)-traced cells co-expressing Glucagon (red, left panel), Somatostatin (red, middle panel) and Insulin (red, right panel) with Ppy (cyan) at P30. Scale bar: 20 μm (10 μm in insets). Region of the pancreas: Ventral. Source data are provided as Source Data file (Supplementary Table e).

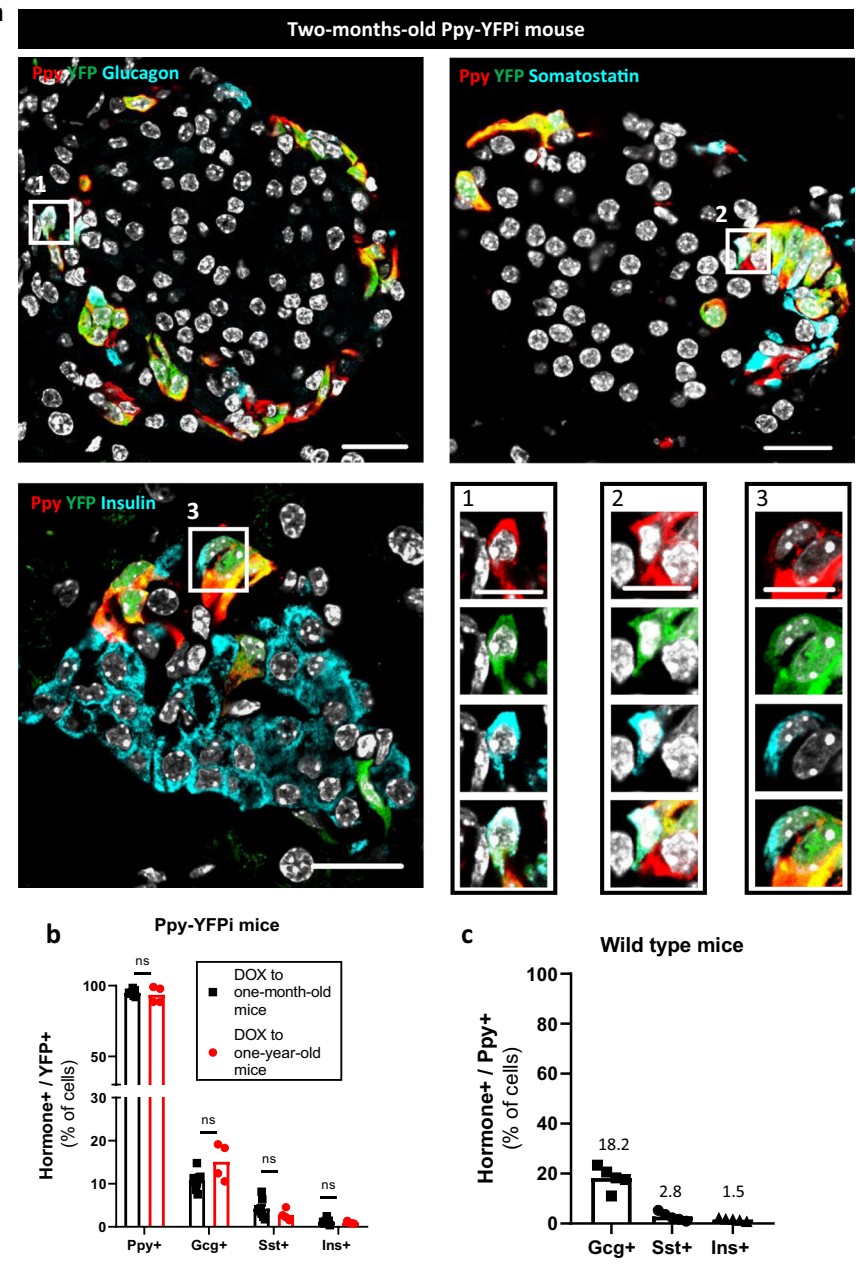

**Fig. 3 Bihormonal islet cells co-express Ppy and another hormone in adult mice. a** Immunofluorescence on pancreatic sections from 2-months-old Ppy-YFPi mice stained with Ppy (red) and YFP (green) in combination with: Glucagon (cyan, top left), Somatostatin (cyan, top right) or Insulin (cyan, bottom left). Bihormonal Ppy-Gcg (1), Ppy-Sst (2) and Ppy-Ins (3) cells are detected (bottom right). Scale bars: 20 μm or 10 μm (insets 1, 2, 3). **b** In young Ppy-YFPi mice ($n = 8$ mice), 94.6%, 10.7%, 4.22% and 1.16% of YFP-traced cells are Ppy+, Gcg+, Sst+ and Ins+, respectively (5411 YFP+Ppy+, 610 YFP+Gcg+, 225 YFP+Sst+ and 81 YFP+Ins+ out of 5714 YFP+ cells scored). In aged Ppy-YFPi mice ($n = 4$ mice), 93.4%, 19.0%, 2.8% and 1.0% of YFP-traced cells are Ppy+, Gcg+, Sst+ and Ins+, respectively (2751 YFP+Ppy+, 418 YFP+Gcg+, 81 YFP+Sst+ and 27 YFP+Ins+ out of 5964 YFP+ cells scored). Two-tailed Mann–Whitney test (ns $p > 0.05$; *$p \leq 0.05$; **$p \leq 0.01$; ***$p \leq 0.001$). **c** In adult wild-type mice ($n = 5$ mice), 18.2%, 2.77% and 1.54% of Ppy+ cells are Gcg+, Sst+ and Ins+, respectively (535 YFP+Gcg+, 73 YFP+Sst+ and 43 YFP+Ins+ out of 2884 Ppy+ cells scored). Data are presented as mean values ± s.e.m. Region of the pancreas: Ventral. See Source data are provided as Source Data file (Supplementary Table f).

Ppy with Gcg, Sst and Ins was also observed in comparable proportions in non-transgenic wild-type mice (Fig. 3c, Supplementary Fig. 5; Source Data f). These observations show that a significant proportion of the adult γ-cell population is bihormonal and this remains constant even in aged mice.

**Characterization of the transcriptional identity of monohormonal Ppy+ cells.** To characterize the transcriptomic profile

and, therefore, the cell identity of monohormonal *Ppy*-expressing γ-cells, we performed single-cell RNA-sequencing (scRNA-seq) on islet cells isolated from DOX-treated adult Ppy-YFPi mice (DOX at 1 month; analysis at 2 months) and sorted by FACS (Fig. 4a, Supplementary Fig. 6; YFP+ and YFP- fractions were collected). First, we removed putative doublet cells from our dataset using two independent tools, DoubletFinder[42] and Scrublet[43]. Doing this, we accept the risk of eliminating actual bihormonal cells having a transitional hybrid (mixed)

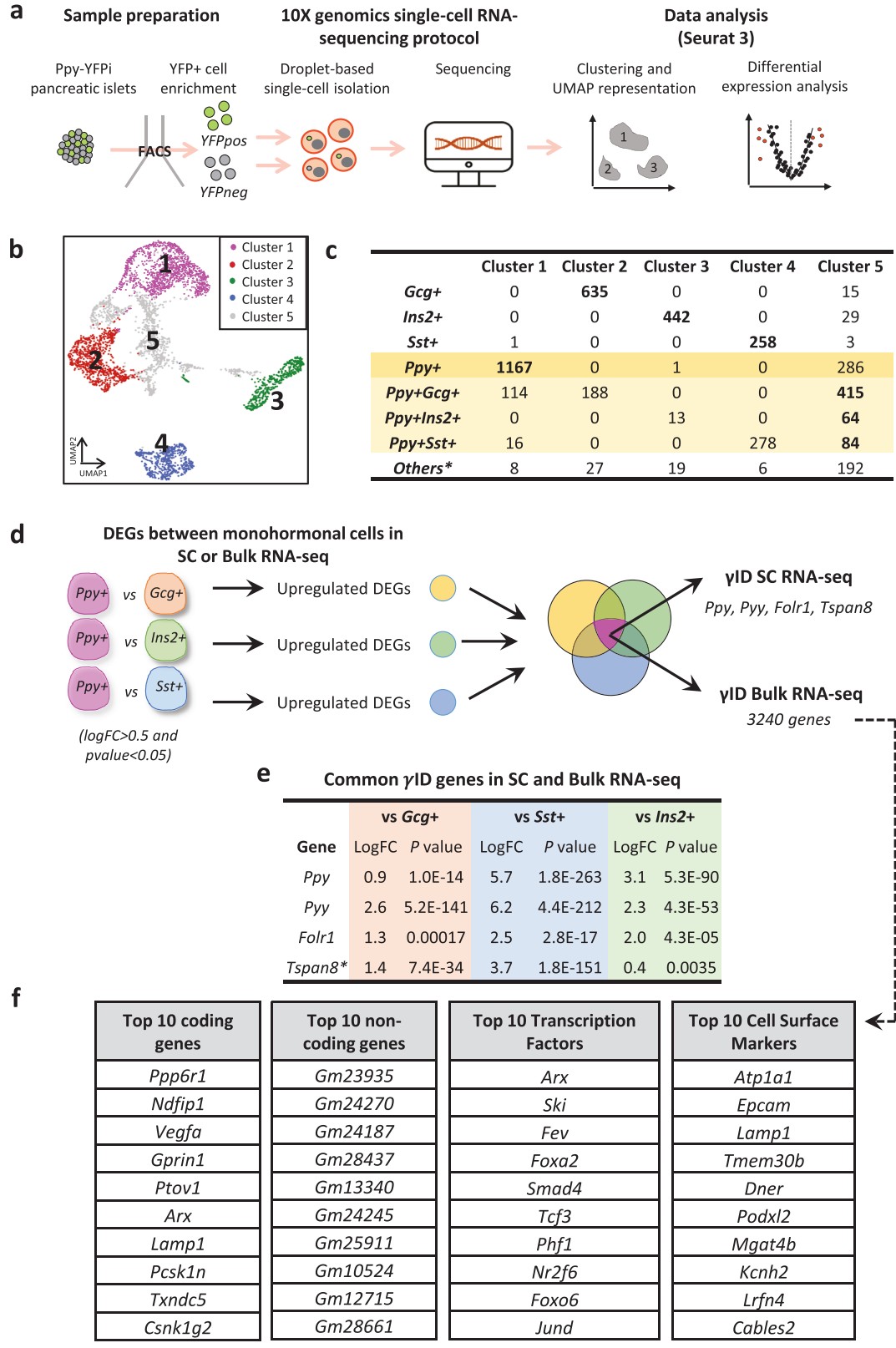

transcriptional profile. After doublet removal, we defined five populations based on unsupervised clustering (Cluster 1–5; Fig. 4b). Each endocrine cell-type cluster was identified using hormone expression: $Gcg^+$ (Cluster 2), $Ins2^+$ (Cluster 3), $Ppy^+$ (Cluster 1), $Sst^+$ (Cluster 4) and $Ppy^+$ bihormonal (Cluster 5) (Fig. 4b, c; Supplementary Fig. 7; see methods). Cells surpassing the mathematically-defined threshold of expression of two

hormones were defined as bihormonal (Supplementary Fig. 7; Source Data g). UMAP dimensional reduction showed that $Ins2$-, $Gcg$-, $Sst$- and $Ppy$-expressing cells were distributed in four main clouds with overlapping regions mainly where $Ppy$ expression is located (Supplementary Fig. 7). Of note, the distribution of the $YFP^+$ cell fraction tightly correlated with the $Ppy^+$ cell localization (Supplementary Fig. 7). Only the hormone-expressing cells

**Fig. 4 Determination of γ-cell identity using single-cell RNA-seq and bulk RNA-seq. a** Experimental design of Ppy-YFPi single-cell RNA-sequencing data. The experiment was performed twice, using 3 and 4 mice, respectively. **b** Clustering analysis (UMAP visualization) of single-cell transcriptomes of Ppy-YFPi islet endocrine cells. **c** Islet endocrine cell distribution in the clustering analysis (**b**). *Ppy-expressing* cells in the table are labeled in yellow. **d** Schematic representation of the strategy to obtain cell-type ID genes in single-cell and bulk RNA-seq. 4 and 3240 γID genes were identified in single-cell and bulk RNA-seq, respectively. **e** LogFold Change (logFC) and P values of the single cell RNA-seq γID genes in the bulk RNA-seq dataset. **f** Table of the top differentially expressed coding genes, non-coding genes, transcription factors and cell surface markers in the γID list obtained with bulk RNA-seq. The top DEGs are ranked based on Z-score. *Tspan8 is not included in the bulk RNA-seq γID gene list because its logFC is lower than 0.5 in the comparison of *Ppy+* vs *Ins2+* cells. Source data are provided as Source Data file (Supplementary Tables g–i).

contained in their assigned cell-type cluster were considered for further analysis (i.e., only the *Gcg*-expressing cells located in Cluster 2 were studied while *Gcg+* cells present in other clusters were excluded from the analysis as these latter transcriptionally resemble to other islet cells, but express *Gcg*; Fig. 4c).

To characterize monohormonal *Ppy*-expressing cells, we performed a pair-wise differential expression analysis between γ-cells and each of the three other monohormonal islet cell-types. The common upregulated genes in each of the endocrine cell comparisons (*Ppy+* vs *Gcg+*, *Ppy+* vs *Sst+* and *Ppy+* vs *Ins2+*) defined the monohormonal γ-cell identity (ID; Fig. 4d; Source Data h). We obtained 4 γ-ID genes, including *Ppy* and *Pyy*, but also *Tspan8 and Folr1*. α-, β- and δ-cell ID gene lists were also generated: 82 α-ID genes, 134 β-ID genes and 9 δ-ID genes were identified and they comprise characteristic cell markers such as *Gcg, Ttr* and *Mafb* in the α-specific genes; *Ins1, Ins2* and *Nkx6-1* in the β-specific genes; *Sst* and *Hhex* in the δ-specific genes (Source Data h). As single cell transcriptomics did not provide the necessary complexity to perform deep profiling, we next made use of bulk RNA sequencing of Ppy-expressing cells to more deeply profile these cells (see methods). In this dataset, 3240 γ-ID genes were obtained, including *Ppy, Pyy* and *Folr1* (Fig. 4d, e; Source Data i). Despite *Tspan8* expression was upregulated in *Ppy*-expressing cells, it did not pass the logFC threshold when *Ppy+* cells were compared to *Ins2+* cells (Fig. 4e). Within the bulk RNA-seq γ-ID, 195 transcription factors such as *Arx, Ski* and *Fev*, and 214 cell surface markers such as the Na+/K+ transporter subunit α-1 (*Atp1a1*) and the Epithelial cell adhesion molecule (*Epcam*) were identified (Fig. 4f).

*Ppy*-expressing γ-cells were also studied using Ingenuity Pathway Analysis (IPA) to better define their functional identity. To maximize the robustness of our analysis, DEGs between γ-cells and each of the other islet cell-types were calculated in the two independent islet RNA-seq datasets: the mouse single-cell RNA-seq and the mouse bulk RNA-seq. This resulted in hundreds or thousands of DEGs (Source Data j) for each of the three islet cell comparisons, which were then uploaded into IPA software. Modulated pathways were only considered if showing the very same regulation in both datasets (i.e., either upregulated or downregulated in both datasets; Source Data k). We identified 7 differentially-modulated γ-cell pathways compared to α-cells. Among these pathways, PI3K/AKT and Sirtuin Signaling Pathway were inhibited and activated, respectively. 46 pathways were modulated in γ-cells as compared to β-cells, including inhibition of the PI3K/AKT and AMPK pathways in γ-cells. Finally, 20 pathways were significantly regulated in γ-cells compared to δ-cells, including EIF2 activation (Source Data k). These pathways along with the ID genes define the functional signature of mouse γ-cells.

**Bihormonal Ppy+ cells express identity genes of α-, δ- or β-cells.** Next, we analyzed the identity of the mouse bihormonal *Ppy +*-cells as compared to the different monohormonal islet cell-types. By applying clustering analysis, we have in silico isolated

the genuine hybrid bihormonal cells (black dots; Fig. 5a–c) from the bihormonal cells located in monohormonal cell clusters (colored dots; Fig. 5a–c). These latter cells were excluded as they transcriptionally resemble to monohormonal α-, β- and δ-cells, but express *Ppy*. *Ppy+Gcg+*, Ppy+Sst+ and Ppy+Ins2+ bihormonal cells represent 23.9%, 4.8% and 3.6% of the total *Ppy+* cell population, respectively (Fig. 5d, Source Data g). Although present in higher proportion in the single-cell dataset, the abundance of each bihormonal *Ppy+* subpopulation correlated with our observations at the protein level (Fig. 3 and Supplementary Fig. 5). Interestingly, most of the bihormonal *Ppy+Gcg+*, *Ppy+Sst +* and *Ppy+Ins2+* cells were scattered in UMAP in between each of the corresponding monohormonal populations (Fig. 5a–c), suggesting that these cells have a mixed transcriptomic signature. To test for this, we assessed how many of the α-, δ- and β-ID genes (Fig. 4 and Source Data i) were expressed in *Ppy+Gcg+*, *Ppy +Sst+* and *Ppy+Ins2+* bihormonal cells, respectively. Out of the 83 α-cell ID genes, 77 (94%) were present in the 199 DEGs detected in bihormonal *Ppy+Gcg+* cells compared to monohormonal *Ppy+* cells. Similarly, 6 (67%) out of the 9 δ-cell ID genes were allocated in the list of upregulated genes in *Ppy+Sst+* cells. Finally, 35 (26%) out of the 134 β-cell ID genes were shared with the upregulated genes in *Ppy+Ins2+* cells (Fig. 5e, f, Source Data l). Interestingly some markers such as *Iapp* and *Chga* were consistently enriched in the 3 different *Ppy+* bihormonal cell types (Source Data m). Enrichment of the *Iapp* and *Chga* in the *Ppy+Gcg+* and *Ppy+Ins2+* populations was confirmed at protein level in islets of two independent wild-type mice (49.6% and 67.9% of Ppy+Gcg+ bihormonal cells vs 18.4% and 17.9% of the monohormonal Ppy+ cells contain Iapp and Chga, respectively; 87.4% and 93.8% of the Ppy+Ins+ bihormonal cells vs 8.1% and 28.1% of the monohormonal Ppy+ contain Iapp and Chga, respectively; minimum islets scored = 22; Supplementary Fig. 8 and Source Data m). These results indicate that *Ppy+* bihormonal cells comprise a significant fraction of typical identity genes with monohormonal α-, δ- and β-cells, thus confirming a hybrid transcriptomic profile and not just simple dysregulation of hormone genes.

**Human islets also contain bihormonal γ-cells.** We next assessed whether bihormonal PPY-expressing γ-cells are also present in the human islet. We recently generated human scRNA-seq dataset from three independent non-diabetic human donors (van Gurp et al, submitted). Islets were sorted by FACS using an antibody-based protocol described previously[44–46], allowing for γ- and δ-cell enrichment (Fig. 6c, Source Data n). Doublet removal and islet cell type identification were performed as described above. Cell-type-specific ID gene lists were calculated using the differentially expressed genes (DEGs) in γ-cells, as compared to the three other islet cell-types in conjunction. A total of 24 human γ-cell ID genes were identified (van Gurp et al, submitted). In order to assess the similarities between mouse and human *PPY*-expressing cells, we intersected the upregulated DEGs in *PPY+* cells compared to *GCG+*, *SST+* and *INS+*

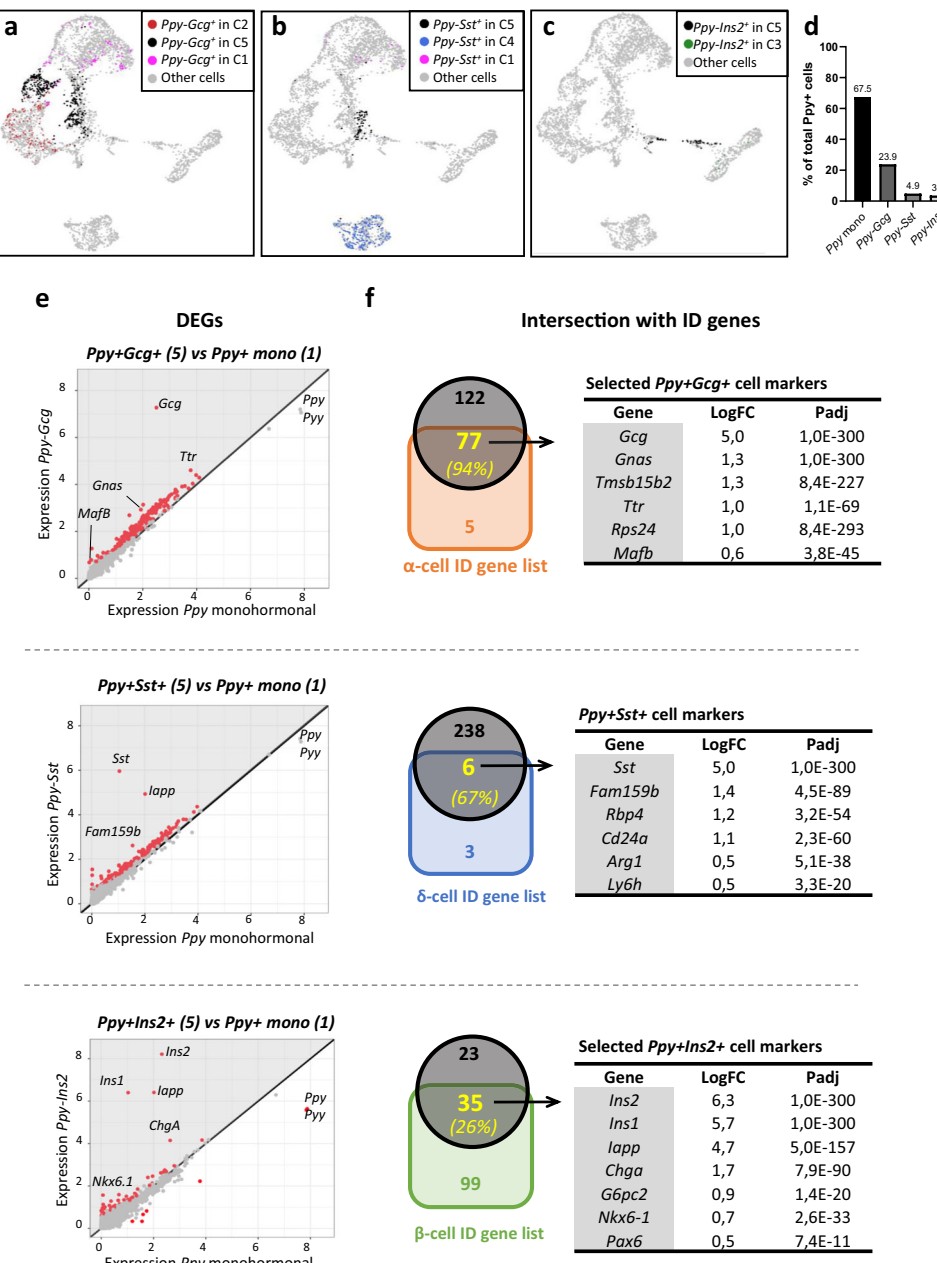

**Fig. 5 Bihormonal γ-cells share identity genes with α-, δ- or β-cells.** UMAP localization of the bihormonal Ppy+ cells in the different clusters: *Ppy+Gcg+* (**a**), *Ppy+Sst+* (**b**) and *Ppy+Ins2+* (**c**). Ppy+ bihormonal cells located outside of Cluster 5 were excluded as they transcriptionally resemble to monohormonal α-, β- and δ-cells, but express *Ppy*. **d** Proportion of *Ppy+* cells expressing other islet hormones (1167 *Ppy+*, 415 *Ppy+Gcg+*, 84 *Ppy+Sst+* and 64 *Ppy+Ins2+* cells). **e** Expression plot of differentially expressed genes (DEGs) between bihormonal *Ppy+Gcg+* (top), *Ppy+Sst+* (middle) and *Ppy+Ins+* (bottom) in cluster 5 and monohormonal *Ppy*-expressing cells in cluster 1. Each dot represents one gene. Red dots represent the genes with *P* value < 0.05. Gene names label the top DEGs. Gray area indicates the upregulated genes in each bihormonal population. Differential expression was calculated using a negative binomial generalized linear model in a pairwise manner between populations, taking along the number of UMIs and the number of genes as variables to regress. *P* values (in panel **f** and Source Data l) are Bonferroni corrected based on the total number of genes in each dataset. **f** Intersection of the upregulated genes in *Ppy+Gcg+* (top), *Ppy+Sst+* (middle) and *Ppy+Ins+* (bottom) from **e**, with mouse α-, δ- and β-cell ID gene lists. Important markers of the adult α-cells (77 out of 82), δ-cells (6 out of 9) and β-cells (35 out of 134) are shared with the bihormonal *Ppy*-expressing cells. Source data are provided as Source Data file (Supplementary Table l).

independently in the three RNA-seq datasets: mouse single cell RNA-seq, mouse bulk RNA-seq and human single cell RNA-seq (Supplementary Fig. 9A; Source Data o). *PPY, SCG2, TTR* and *GC* were shared in both species between the three RNA-seq datasets (Supplementary Fig. 9A). The restricted number of genes shared between the 3 datasets is likely due to some inter-species variation as previously documented for some genes, like *PYY* expression[47];

but also to variation in resolution in the single cell RNA-sequencing techniques. This list of γ-cells genes shared between both species can be expanded to 24 genes when comparing upregulated genes between the human single cell RNA-seq and either the bulk or the single cell mouse datasets (Supplementary Fig. 9A; Source Data p). Using IPA, we defined the pathways modulated in human γ-cells and intersected with those observed

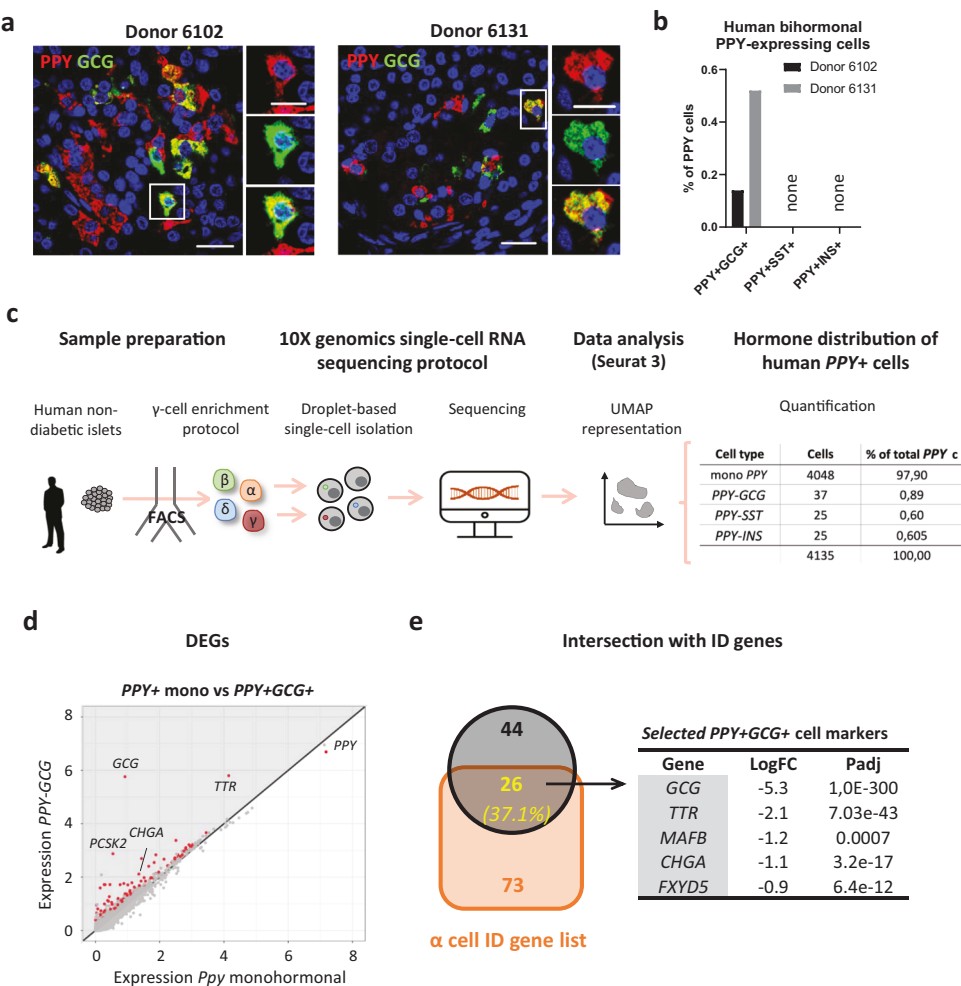

**Fig. 6 Human islets contain bihormonal cells co-expressing PPY and another hormone. a** Bihormonal PPY+GCG+ cells could be detected by immunofluorescence (insulin, red; glucagon, green) in human islets from two donors. Scale bar: 20 μm (islet) or 10 μm (cell). **b** Quantification of bihormonal PPY+ cells in human islet sections (n = 2 human donors; in donor 6102, 2 PPY+GCG+ were detected in 1395 PPY+ cells; in donor 6131, 6 PPY+GCG+ cells were detected in 1154 PPY+ cells). **c** Experimental protocol and analysis of human islets single-cell RNA-sequencing data. The experiment was performed once with islets from three independent donors (n = 3 human donors). 4135 PPY-expressing cells were obtained after the analysis. Among them, bihormonal PPY-expressing cells could be detected at transcriptomic level: 37 PPY+GCG+ cells, 25 PPY+SST+ cells and 25 PPY+INS+ cells. **d** Expression plot of differentially expressed genes (DEGs) in bihormonal PPY+GCG+ relative to monohormonal PPY-expressing cells. Each dot represents one gene. Red dots represent the genes with P value < 0.05. Gene names label the top DEGs. Gray area indicates the upregulated genes in the bihormonal population. Differential expression was calculated using a negative binomial generalized linear model in a pairwise manner between populations, taking along the number of UMIs and the number of genes as variables to regress. P values (in panel **e** and Source Data s) are Bonferroni corrected based on the total number of genes in each dataset. **e** Intersection of the upregulated genes in PPY+GCG+ from **d**, with human α-cell ID gene list. 26 out of 70 PPY+GCG+ -enriched genes were identified as important markers for adult α-cells. Source data are provided as Source Data file (Supplementary Tables n, r, s).

in mouse γ-cells (Supplementary Fig. 9B–D; Source Data q). 3 and 15 pathways exhibit the very same modulation in mouse and human γ-cells, as compared to α-cells and β-cells, respectively (Supplementary Fig. 9C, D). No common pathway between both species was identified when comparing γ- and δ-cells (Source Data q).

Although with lower frequencies, the three types of bihormonal PPY+ cells were also detected in human PPY-expressing cells: 37 PPY-GCG+, 25 PPY-SST+ and 25 PPY-INS+ cells (Fig. 6c; Source Data n). PPY-GCG coexpressing cells were also detected at the protein level in pancreatic sections from the two non-diabetic donors analyzed (Fig. 6a, b, Source Data r). Deeper analysis of the PPY+ bihormonal cells by scRNA-seq, identified 70 upregulated genes in PPY-GCG bihormonal compared to monohormonal PPY cells (including the Ppy+ bihormonal-cell marker CHGA), out of which 26 (37.1%) were common with the 99 human α-cell ID

genes (van Gurp et al, submitted). This includes functional α-cell markers such as GCG, TTR and MAFB (Fig. 6d, e; Source Data s). Similarly, PPY-SST and PPY-INS bihormonal cells shared some typical markers found in human δ- and β-cells, respectively (Supplementary Fig. 10; Source Data s).

These results indicate that bihormonal PPY-expressing cells also exist in adult human islets at both protein and mRNA level. These cells share common markers with the corresponding monohormonal cell types as we observed in mice.

**Ppy gene inactivation and cell ablation do not affect blood glucose levels or body weight.** Previous studies have suggested a role for Ppy on appetite suppression[22–25]. Here, we assessed the consequence of constitutive Ppy inactivation (gene knockout, "KO") on blood glucose regulation in Ppy-rTA homozygous

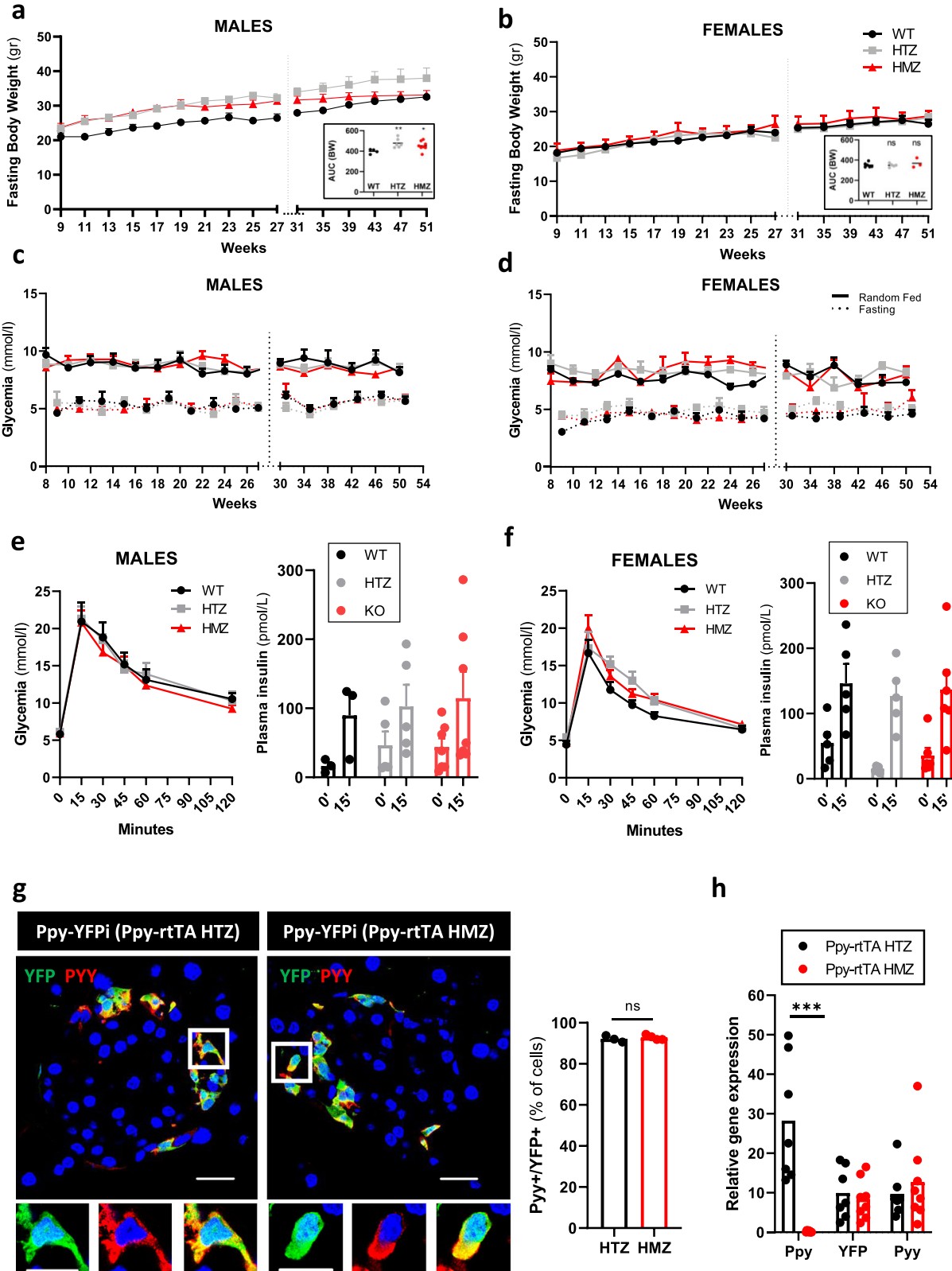

(HMZ) mice. No major alterations in body weight, glycemia and glucose tolerance (intraperitoneal glucose tolerance tests, "i.p. GTT") were observed in Ppy-rtTA HMZ mice of both genders when compared with wild-type (WT) and Ppy-rtTA heterozygous (HTZ) mice (Fig. 7a–f; Source Data t). In addition, no differences in the glucose-stimulated insulin secretion capacity were detected between Ppy KO mice of both genders and wild-type (WT) or

Ppy-rtTA HTZ mice (Fig. 7e, f; Source Data t). Because functional redundancy between NPY family members has been previously reported[48,49], we assessed whether Pyy protein is expressed in γ-cells, as it is suggested by mRNA analyses (Fig. 7g and[47]; Source Data u). About 90% of the YFP+ population co-expressed Pyy in Ppy-rtTA HTZ mice (Fig. 7g; Source Data u). The expression level of *Pyy* in the islet was not impacted upon *Ppy* gene KO

**Fig. 7 Ppy inactivation has no impact on body weight and glycemia.** Fasting body weight curves in wild-type (black), Ppy-rtTA HTZ (gry) and Ppy-rtTA HMZ (red) males (**a**) and females (**b**) mice. Area under de curve (AUC) of each genotype is shown in the bottom right panel. Glycemia curves in wild-type (black), Ppy-rtTA HTZ (gray) and Ppy-rtTA HMZ (red) males (**c**) and females (**d**) mice. Solid line represents random fed glycemia values, dashed line represents fasting glycemia values. Body weight and glycemia values were obtained from the same mice. Males: WT, $n = 4$ mice; HTZ, $n = 6$ mice; HMZ; $n = 8$ mice. Females: WT, $n = 6$ mice; HTZ, $n = 5$ mice; HMZ, $n = 3$ mice. Intraperitoneal glucose tolerance test to 1-year-old wild-type (black), Ppy-rtTA HTZ (gray) and Ppy-rtTA HMZ (red) males (**e**, left panel) and females (**f**, left panel) mice. Glucose-stimulated insulin secretion in vivo to 1-year-old wild-type (black), Ppy-rtTA HTZ (gray) and Ppy-rtTA HMZ (red) males (**e**, right panel) and females (**f**, right panel) mice. Males: WT, $n = 3$ mice; HTZ, $n = 6$ mice; HMZ; $n = 8$ mice. Females: WT, $n = 6$ mice; HTZ, $n = 5$ mice; HMZ, $n = 6$ mice. **g** Most of the YFP-labeled γ-cells express also Pyy. Percentage of YFP-tracked Ppy-expressing cells that co-express Pyy in Ppy-rtTA HTZ ($n = 3$ mice; 1522 Pyy+YFP+ out of 1653 YFP+ cells scored) or Ppy-rtTA HMZ ($n = 4$ mice; 1973 Pyy+YFP+ out of 2118 YFP+ cells scored) mice. **h** qPCR of *Ppy*, *YFP* and *Pyy* in FAC-sorted YFP+ cells of Ppy-rtTA HTZ ($n = 7$ mice) and HMZ ($n = 8$ mice) mice. Data are shown as normalized ct values relative to β-actin. Two-tailed Mann–Whitney test (ns $p > 0.05$; *$p \leq 0.05$; **$p \leq 0.01$; ***$p \leq 0.001$), Ppy expression in Ppy-rtTA HTZ versus HMZ: $P$ value = 0.0006; YFP expression Ppy-rtTA HTZ versus HMZ: $P$ value = 0.6943; Ppy expression Ppy-rtTA HTZ versus HMZ: $P$ value = 0.6126. Only the significant statistical differences are indicated in the figure. Region of the pancreas: Ventral. Scale bar: 20 μm (islet) or 10 μm (cell). Data are presented as mean values ± s.e.m. Source data are provided as Source Data file (Supplementary Tables t, u).

(Fig. 7h; Source Data u). While it cannot be excluded that Pyy may compensate for the lack of Ppy action, our results suggest that Ppy absence is well tolerated and has no impact on body weight or blood glucose regulation, at least under basal conditions.

As Ppy hormone inactivation did not impact on body weight evolution nor blood glucose regulation, we generated a knock-in mouse that enabled us to selectively ablate all Ppy-expressing cells. To do this, we replaced the coding sequence of the *Ppy* gene by the diphtheria toxin receptor (DTR; Supplementary Fig. 11A, B). In these KI mice, diphtheria toxin (DT) administration efficiently induced more than 99% γ-cell ablation 15 days and 3 months post-injection (Supplementary Fig. 11C; 0.004 Ppy+ cells per islet section were identified 15 days post-ablation; 0.1 Ppy+ cells per islet section were identified 3 months post-ablation; Source Data v). Similar to the *Ppy* inactivation mouse model, no alterations in body weight, glycemia and i.p.GTT were detected in mice lacking γ-cells (Supplementary Fig. 11D–I; Source Data v).

Although we cannot completely exclude that the remaining 1% γ-cells may release undefined factors that could be involved in blood glucose regulation, these results combined suggest that Ppy and the γ-cells are dispensable in basal conditions for blood glucose regulation.

**Ppy-expressing cells efficiently engage insulin production.** We previously reported that human primary α- and γ-cells can be reprogrammed to glucose-dependent insulin secretion when Pdx1 and MafA expression is induced[44]. In mice, adult α-cells also exhibit cell plasticity by starting insulin production upon β-cell destruction[10]. We thus explored whether γ-cells can spontaneously reprogram to insulin production in diabetic mice. Ppy-YFPi mice were crossed with RIP-DTR animals, in which the diphtheria toxin receptor (DTR) is borne on the surface of β-cells[10] (Fig. 8a). β-cell loss was induced in DOX-treated Ppy-YFPi;RIP-DTR mice by DT administration (Fig. 8b). DT-treated mice became severely hyperglycemic one-week after β-cell ablation (Fig. 8C). The strong decrease in YFP-labeled Ins+ cells 5 days after DT (black vs green dots in Fig. 8e and d; Source Data w) indicates that pre-existing bihormonal Ppy+Ins+ cells were efficiently ablated together with β-cells upon DT. ~40% of the Ins+ cells were YFP-labeled 6 weeks post-DT, indicating that these cells are reprogrammed γ-cells that have started insulin production (Fig. 8f). Nearly all converted γ-cells (95.3% ± 0.1; $n = 4$; 103 Ins+YFP+Ppy- out of 108 Ins+YFP+ cells scored) did not have detectable levels of Ppy (inset in panel Fig. 8g; Source Data w). In total, only a small fraction (3%) of the adult γ-cell population had spontaneously engaged insulin production after

β-cell loss (Fig. 8e, f; Source Data w), as previously reported for α-cell conversion in diabetic mice[10]. Interestingly, up to 40% of YFP-labeled γ-cells could be convinced to produce insulin when combined with ectopic Pdx1 expression in γ-cells and DT-induced β-cell loss (Supplementary Fig. 12; Source Data x). Together, these results indicate that mouse γ-cells exhibit a functional cell plasticity like α- and δ-cells, as we previously showed with human primary PPY-expressing cells.

## Discussion

Using transgenic mouse lines for conditional cell lineage tracing and ablation, in combination with human islet flow cytometry sorting (FACS) techniques, we provide here a deep characterization of the genetic identity of the very elusive PPY-expressing γ-cell at the mRNA and protein levels. We have uncovered a seemingly distinct and significantly abundant population of bihormonal γ-cells in the adult mouse and human islet. We also determined that *Ppy* gene inactivation, or massive γ-cell ablation, has little or no impact on blood glucose homeostasis under basal conditions. Finally, our results show that γ-cells can also reprogram after massive β-cell ablation and, along with our previous reports[10,11] show that in essence all non-β-cells have the capacity to produce insulin in vivo under the appropriate stress conditions, collectively contributing to β-cell regeneration.

The current consensus is that under physiological conditions essentially all adult islet cells are monohormonal, as dysregulation of hormone expression could lead to detrimental secretory dynamics. Yet, we found that about 15% of adult mouse γ-cells express an additional islet hormone at the protein level, as well as additional identity markers shared with other islet cell-types at the mRNA level. For instance, *Ppy+Gcg+* cells expressed typical α-cell markers like *Ttr* and *MafB*. *Ppy+Sst+* and *Ppy+Ins2+* cells also expressed key δ- and β-cell markers (*Hhex* in *Ppy+Sst+* cells and *Nkx6.1* in *Ppy+Ins+* cells, for instance). These observations indicate that the different *Ppy+* bihormonal cells display hybrid transcriptional profiles (Figs. 4 and 5). Human γ-cells were also bihormonal, yet less frequently than in mice.

Future research should focus on the precise origin and functional role of these bihormonal *Ppy+* cells, with new methodology needed to determine whether their bihormonal status is permanent or dynamic. We and others have reported the presence of *Ppy+* bihormonal cells in developing mouse and human pancreas[38,50,51]. Therefore, bihormonal cells arise during development and are maintained, at least in part, after birth. In addition, a minor fraction of the postnatal bihormonal cells might originate from *Ppy+* monohormonal γ-cells, or inversely, from α-, β- or δ-cells that initiate *Ppy+* expression. At this point, more

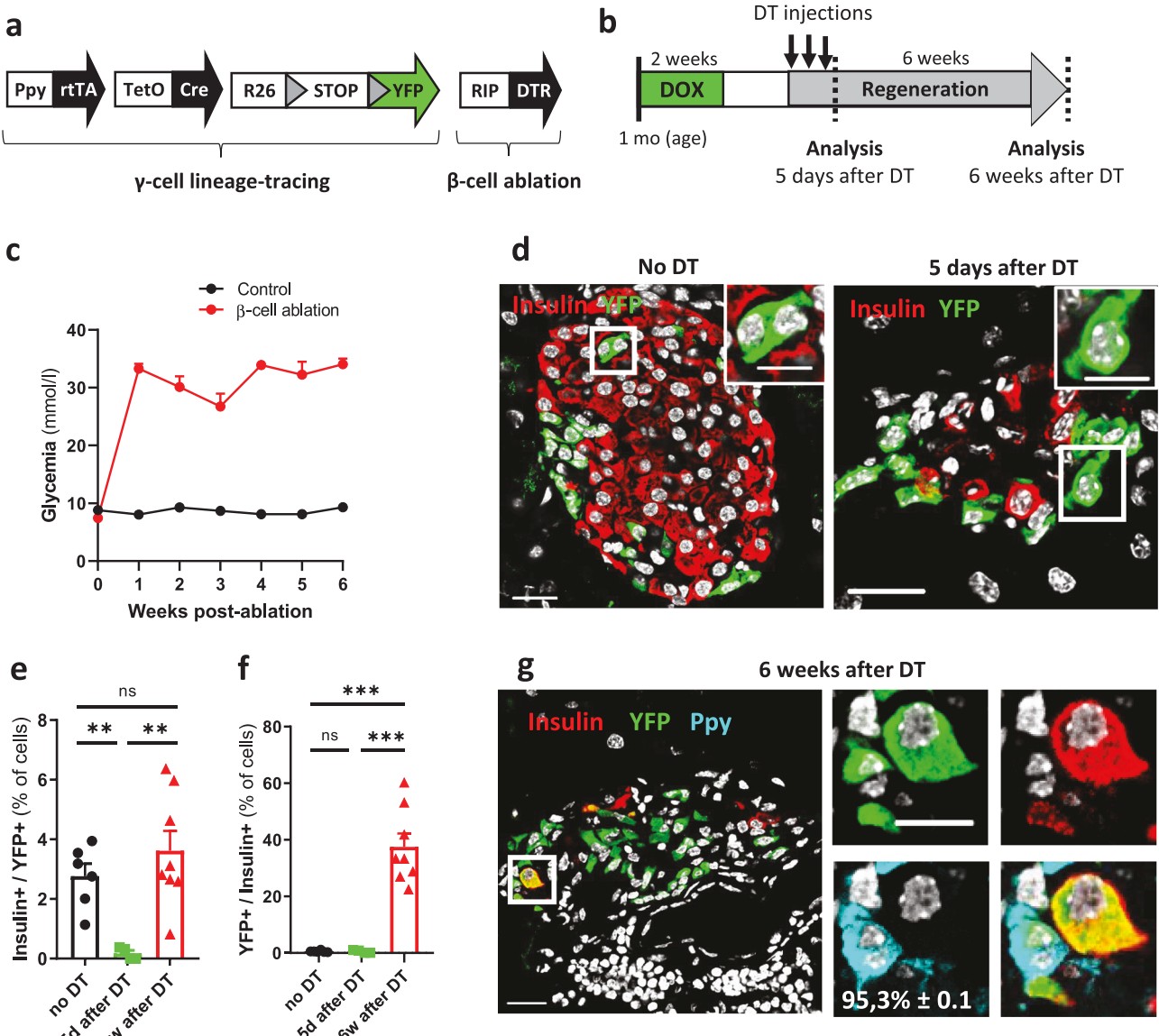

**Fig. 8 Murine γ-cells engage insulin production after massive β-cell loss. a** Transgenes required for tracing γ-cells in combination with β-cell ablation. **b** Experimental design. **c** β-cell ablated mice ($n = 7$ mice) become severely diabetic one week after β-cell death compared to control mice ($n = 6$ mice). Insulin implants were administrated one week after DT-mediated β-cell death. **d** Adult YFP-labeled cells can engage insulin expression after diphtheria toxin (DT)-mediated β-cell loss. Immunofluorescence of insulin (red) and YFP (green) in control ("No DT") and 5 days or 6 weeks after DT injection. Almost no YFP-labeled cells contained insulin 5 days after DT (0.1%; $n = 4$ mice). Scale bars: 20 μm or 10 μm (insets). **e** Proportion of YFP-labeled cells that start insulin production after β-cell ablation. Two-tailed Mann–Whitney test, no DT versus 5d after DT: $P$ value $= 0.0095$; 5d after DT versus 6w after DT: $P$ value $= 0.004$; no DT versus 6w after DT: $P$ value $= 0.662$. Data are presented as mean values ± s.e.m. **f** Quantification of insulin-producing cells labeled with YFP after DT-mediated β-cell loss. 6 weeks after β-cell loss, 40% of the insulin-containing cells found are reprogrammed γ-cells. Data are presented as mean values ± s.e.m. Two-tailed Mann–Whitney test, no DT versus 5d after DT: $P$ value $= 0.709$; 5d after DT versus 6w after DT: $P$ value $= 0.002$; no DT versus 6w after DT: $P$ value $= 0.0007$. no DT, $n = 6$ mice, 126 Ins+YFP+ out of 6362 YFP+ cells scored; 5d after DT, $n = 4$ mice, 6 Ins+YFP+ out of 2826 YFP+ cells scored; 6w after DT, $n = 8$ mice, 209 Ins+YFP+ out of 5477 YFP+ cells scored. **g** Nearly all converted γ-cells do not have detectable levels of Ppy (95.3% ± 0.1; $n = 4$ mice; 103 Ins+YFP+Ppy- out of 108 Ins+YFP+ cells scored). Insulin, red; YFP, green; Ppy, cyan. Scale bars: 20 μm or 10 μm (insets). Region of the pancreas: Ventral. Source data are provided as Source Data file (Supplementary Table w).

sophisticated bihormonal-cell tracing tools would be required to better study their origin and transitional dynamics.

Our results also provide further insight into the requisite of Ppy and γ-cells in blood glucose regulation. Despite the reported role of Ppy controlling appetite, here and in previous reports[27], constitutive *Ppy* gene inactivation did not alter mouse body weight. Furthermore, glycemic monitoring did not reveal any affection on blood glucose homeostasis and glucose-stimulated insulin secretion (Fig. 7) under the conditions analyzed here. Of

note, these results differ from the observations made in several *Npyr* knockout mouse models[52,53], where the hypophagic effect of Ppy is absent. Inactivation of an Npy receptor affects the signaling pathways of all Npy peptides (Npy, Ppy and Pyy)[54,55], resulting in a robust phenotype (reduced food intake and body weight). By contrast, *Ppy* inactivation affects Ppy signaling pathway specifically. It is possible that the appropriate experimental conditions have not been tested, as Ppy might have more of a fine-tuning role in response to significant islet secretory

dysregulation due to severe stresses, like viral infections or prolonged fasting. Interestingly, previous reports have shown that male and females Ppy-deficient mice treated with high-fat diet display no change in body weight gain or body composition compared to wild-type littermates[27]. This observation is in line with the absence of phenotype in mice in which the hormone or the γ-cells were inactivated and ablated, respectively (Fig. 7 and Supplementary Fig. 11).

Alternatively, this lack of phenotype might be due to compensatory mechanisms that ensure residual Ppy signaling in Ppy KO mice. Several studies revealed that the Ppy gene originated from a duplication of the Pyy gene, and both genes are only 10 kb apart in mouse and human genomes[56,57], which might be compatible with a transcriptional co-regulation. In line with this, we found that mouse γ-cells do express significant levels of Pyy, both in wild-type and Ppy-KO mice (Fig. 7). As both proteins act on the same family of Npy receptors[48,54], Pyy action might compensate the lack of Ppy expression. Yet, γ-cell ablation did not reveal any significant metabolic phenotype, implying that the putative compensatory Pyy effect should be due to the activity of intestinal L-cells, which secrete Pyy[58,59]. The generation of a conditional double Ppy-Pyy KO mouse would be required to elucidate the function of these Npy family peptidic hormones.

We recently reported that human γ-cells can be efficiently reprogrammed into glucose-responsive insulin-secreting cells upon Pdx1 and MafA overexpression[44]. Here, we confirmed this plasticity in vivo in mouse γ-cells. Like α- and δ-cells[10,11], a small fraction of Ppy-expressing γ-cells (2–3%) spontaneously reprogram to produce insulin after extreme β-cell ablation. Like α- and δ-cells, the proportion of γ-cells that become insulin expressers can also be synergistically increased by forcing Pdx1 expression after β-cell ablation (up to 40%). Intriguingly, unlike α-cells, about half of the Pdx1+ γ-cells seemed refractory to reprogramming. This would be compatible with the heterogeneity observed here, and could indicate that the plasticity potential toward insulin production may be limited to a γ-cell subpopulation.

Systematic probing of cell populations with more sensitive methods will help expand our understanding of the versatile nature of cell identity. Whether cells with hybrid transcriptional character are also present throughout other organs, perhaps acting as functional cell reservoirs to be used in the direst of circumstances, remain open and very interesting questions. It is becoming apparent that cells exist in a continuum of functional and genetic states, which could contribute to how organisms adapt to the ever-changing environment.

## Methods

**Animals.** In order to generate the Ppy-rtTA and Ppy-DTR knock-in mice, two regions of the Ppy locus were targeted with CRISPR: 5′-GGAGAGGCAGCAGTATGCGA-3′ and 5′-TGATTCCCTGCTCTGCGCCC-3′. These sequences comprise the ATG and STOP codon sides of the Ppy locus, respectively. The two targeted sgRNA were cloned into the Cas9-containing pX330 vector[60]. The donor plasmid (pBluescript II SK+) contained the rtTA (Tet-On 3G from Clontech) or DTR[10,61] sequence flanked by 2 kb right homology arm (upstream of the ATG in Ppy locus) and 2 kb left homology arm (downstream of the STOP codon in Ppy locus). All three plasmids (two pX330 containing the sgRNA and the donor plasmid containing the rtTA sequence) were co-injected in one cell embryos. Ppy-rtTA transgene was combined with the previously described TetO-Cre, R26-YFP, RIP-DTR and CAG-STOP-Pdx1[62] transgenes.

Mice were housed in open cages with density varying depending on the size of the cage, in accordance with the Swiss regulation (Cage type S to L, Charles River). Cages were enriched with bedding, nestlet and a mouse house. Temperature and humidity in the housing room was maintained between 20–24 °C and 30–70%, respectively. The day-night cycles were programmed by alternating 12 h day–12 h night. Animals received food and tap water ad libitum.

Male and female mice were used in all experiments. Animals were randomly allocated to control or treatment groups. Because of γ-cell distribution and abundance, only ventral pancreas was analyzed. The study follows all ethical regulations regarding animal experimentation, all experiments were performed under the guidelines of the Direction General de la Santé du Canton de Genève (license numbers: GE/111/17 and GE/121/17).

**Human samples.** All studies involving human samples were approved by ethical committee in University of Geneva. Pancreatic histologic samples were obtained from anonymized deceased patients through the nPOD (Network for Pancreatic Organ Donors with Diabetes), supported by JDRF (Juvenile Diabetes Research Foundation International) at the U. of Florida. Donor information and consent from the donor family were obtained for all nPOD samples.

**Diphtheria toxin, doxycycline, tamoxifen and insulin treatments.** Diphtheria toxin (DT, Sigma) was injected intra-peritoneal (126 ng of DT per injection, on days 0, 3 and 4) to 2-month-old mice. Doxycycline (1 mg/ml; Sigma) was administered in the drinking water for two weeks. Tamoxifen (TAM, Sigma) was diluted (20 mg of TAM in 50 μl 100% ethanol and 950 μl corn oil) and injected intra-peritoneal (2 doses of 5 mg, 2 days apart) to 2-month-old mice. Mice received insulin pellet subcutaneously (Linbit) upon hyperglycemia (>25 mM).

**Islet isolation, FACS and RNA extraction.** Islet isolation and cell sorting using FACS were performed as described in previous work[11,62] using a BD FACSAria II or Moflo Astrios (Beckman Coutler) system. FACSDiva v 8.0.1 (BD Biosciences) software was used for sorting on a FACSAria2, Summit v 6.2 (Beckman Coulter) for sorting on a Moflo Astrios. Kaluza Analysis v 2.0 (Beckman Coulter) software was used for subsequent analysis. Islets were frozen in RLT buffer (Qiagen) with β-Mercaptoethanol and stored at −80 °C before being processed for RNA extraction. RNA extraction from islets was prepared using the Qiagen RNeasy Micro Kit.

**RT-qPCR.** cDNA was generated using the Qiagen QuantiTect Reverse Transcription Kit. Total islets or YFP+ cells of Ppy-rtTA; R26-YFP mice were purified as described above. qPCR reactions were performed using the appropriate primers mixes for each gene as well as the Express SyBr© GreenER kit (Invitrogen #100001652). We used the CorbettRobotics4 robot and the PCR reaction was completed in the CorbettResearch6000 series cycler using a 40 cycles program. Normalization and analysis of the data were done with the RT-PCR analysis_macro v1.1 (from the Genomic Platform, University of Geneva) using two normalization genes (Gapdh and β-actin). Samples were run in triplicate. Primers sequences are shown in Supplementary Table 1.

**Single cell RNA-sequencing.** Islet cells were obtained from two-month-old doxycycline-treated Ppy-rtTA; R26-YFP mice. Islets were isolated and dissociated as described above. Two independent experiments were performed, containing islets from three and four mice, respectively. Single-cell dissociated islets were sorted on a Moflo Astrios (Beckman Coutler) system. Sorted YFP+ and YFP- cells of Ppy-rtTA; R26-YFP were loaded separately for single-cell RNA-seq using the Chromium Controller (10× Genomics) following the Single Cell 3′ Library kit v2 manufacture's protocol. Libraries were sequenced as 100 bp paired-end reads on a HiSeq 40000 platform (Illumina). Reads were demultiplexed and counted using Cell Ranger software pipeline (v2.0.1; 10× Genomics) using Ensembl mm10 build 84 reference genome with the addition of YFP and rtTA transgenic sequences.

Data were loaded as Seurat objects (v3.1.0) in R (version 3.6.1) using R Studio (version 1.0.153) with the Read10X function. Positive and negative fractions from each replicate were combined into a single sparse matrix before being used to generate a Seurat object, with min.features set to 500. Cells were then filtered to contain at least 1000 UMIs and at most 7% mitochondrial genes. Data were normalized using the LogNormalize method, using 10'000 as scale factor. The top 1000 most variable genes were detected using the vst method after which all genes were scaled with use.umi set to TRUE, then principle component analysis (PCA) was performed to find the first 40 PCs.

Next, doublets were detected in each replicate using both DoubletFinder and Scrublet, assuming 2% doublets. In DoubletFinder, pN was set to 0.25 and pK to 0.30 (replicate 1) and 0.06 (replicate 2), according to parameter sweeps on each independent replicate. In Scrublet, the threshold for call_doublets was set to 0.2 for both replicates. Cells designated by either tool as a doublet were removed prior to downstream analysis.

At this point, both datasets were integrated using Seurat FindIntegrationAnchors function, using 20 CC's. The integrated object was scaled with use.umi set to FALSE, the first 50 PCs were calculated and tested using jackstraw (all 50 were significant). Clustering was performed with a resolution of 0.1 after nearest neighbor detection. UMAP dimensional reduction was performed using the uwot method and the 30 nearest neighbors for local approximation. Differential expression between identities was calculated using the FindMarkers function in Seurat, using the negative binomial test on the RNA assay, with experiment replication set as a variable to regress, and a minimum log fold change set to 0.5. Marker genes were interpreted to determine which clusters consisted of endocrine cells, and the final dataset was filtered to contain just these cells. On this subset, PCA and UMAP dimensional reduction were rerun. Then, PCA, clustering and UMAP dimensional reduction were performed with a resolution of 0.8 after nearest neighbor detection.

For each of the four main hormones (*Gcg*, *Ins2*, *Ppy* and *Sst*), density plots were made of their expression distribution of normalized data in the entire dataset. This yielded bimodal plots, in which the local minima between modes could be mathematically determined, using a derivate function based on the *bimodality_amplitude* function of the *modes* package in R. Cells were assigned to be expressing any given hormone if its expression for that hormone was higher than the calculated local minimum value. Based on this method, each cell was assigned to express one or more hormones. The (combination of) expressed hormone(s) was used as an identity for downstream analysis. Only endocrine cells (hormone$^+$ cells) were considered for further analysis. Any monohormonal and bihormonal Ppy-expressing cell not contained in its respective cluster was not included in the subsequent analysis.

Identity genes were generated by comparing mono-hormonal cell types in a pairwise manner (e.g., *Gcg*-expressing vs. *Ins2*-expressing, *Ppy*-expressing and *Sst*-expressing cells). Upregulated genes in bihormonal Ppy-expressing cells were calculated by comparing *Ppy* mono-hormonal cells with either *Ppy-Gcg* bi-hormonal cells, *Ppy-Ins2* bi-hormonal cells or *Ppy-Sst* bi-hormonal cells, calculating differential expression as described above.

**Isolation of human islet cell types**. All studies involving human samples were approved by ethical committee in University of Geneva. Human islets from three independent non-diabetes donors were dissociated and stained with cell surface antibodies as described previously[44–46]. Briefly, labeled cells were sorted on a Moflo Astrios (Beckman Coulter) system. Doublet and dead cells were removed by forward scatter, side scatter, pulse-width parameters and negative staining for DAPI (D1306, Invitrogen). Single viable islet cells were gated in HIC3-2D12 vs. HIC1-2B4 plots and, then, in a CD9 vs. SSC-H plot. Endocrine cell fractions were processed using the Chromium single cell gene expression protocol v3 (10× Genomics).

**Human islet single-cell transcriptome**. Single data from Van Gurp et al. (submitted, under revision) was downloaded from GSE150724 (https://www.ncbi.nlm.nih.gov/geo/query/acc.cgi?acc=GSE150724). Data were used as described in the original manuscript. Doublet cell removal using DoubletFinder and Scrublet was performed as described above. Differential expression was calculated as described above.

**Pathway analysis**. Pathway analysis was performed with ingenuity pathway analysis (IPA, QIAGEN, http://www.qiagen.com/ingenuity). DEGs between monohormonal *Ppy+* vs monohormonal *Gcg+*, *Ins2+* and *Sst+*-expressing cells were used for IPA analysis. DEGs were calculated as described above. IPA was performed using the following settings: expression value type (exp log ratio), reference set (ingenuity knowledge base), relationships to consider (direct and indirect relationships), interaction networks (35 molecules/network and 25 networks/analysis), data source (all), confidence (experimentally observed), species (human and mouse), tissue and cell lines (all), mutations (all).

**Bulk transcriptomics analysis**. Islet cells were obtained from two-month-old doxycycline or tamoxifen-treated Glucagon-rtTA; R26-YFP[10] (Gcg-YFPi), SstCre; R26YFP[11] (Sst-YFPi), RipCreER; R26YFP[63] (RipCreER) and Ppy-rtTA; R26-YFP (Ppy-YFPi) mice. Islets were isolated and dissociated as described above. Cells were purified using a BD FACSAria II or Moflo Astrios (Beckman Coutler) system. RNA was extracted as described above and assessed for quality by Agilent bioanalyzer prior to library generation and sequencing. Libraries preparation, RNA-sequencing and quality controls were performed in the Genomics Core Facility of the University of Geneva. Reverse transcription and cDNA amplification were performed using the SMARTer Ultra Low RNA kit (Clontech). cDNA libraries were prepared using Nextera XT DNA Sample Preparation kit (Illumina) and sequenced on an Illumina HiSeq 2500 (for α-, β- and δ-cells) and HiSeq4000 platform (for γ-cells) with single-end 100-bp reads (for α-, β- and δ-cells) or pair-end 100 bp reads (for γ-cells).

All sequencing data were uploaded to, and aligned on the Galaxy project[64] against Ensembl reference genome GRCm38.p6 (release 100) using STAR version 2.7.2b[65] in 2-pass mapping mode. Aligned data were counted using HTSeq version 0.9.1[66] in union mode. Analyses were performed in a pair-wise manner between YFP-positive samples and alpha cell samples, YFP-positive samples and beta cell samples and YFP-positive samples and delta cell samples using DESeq2 version 1.28.1[67]. For each pairwise comparison, genes were discarded if they had fewer than 5 counts per sample on average. After calculating differential expression between groups, log2 fold changes were shrunken using the normal estimator[67]. Genes were considered to be differentially expressed if the absolute shrunken log2 fold changes were equal to or above 1, and adjusted *p* values were equal to or below 0.05.

γ-ID genes were intersected with a published murine cell surfaceome study[68] to obtain the γ-ID cell surface markers. Similarly, the γ-ID genes were intersected with the orthologues of the published human transcription factor atlas[69] to obtain the γ-ID transcription factors.

**i.p. glucose tolerance test**. Mice were fasted for 16 h before starting the experiments. Intraperitoneal glucose tolerance test (ipGTT) was performed as described[10]. A 20% glucose solution was injected i.p. to fasted mice relative to their body weight. Glycemia was measured before the injection and 15, 30, 45, 60 and 120 min after glucose administration.

**Immunofluorescence**. Cryostat section were 10 μm thick. The primary antibodies used were: guinea pig anti-Pdx1 (1/750; C.W. Wright), guinea pig anti-porcine insulin (1/400; DAKO, A0564), rabbit anti-insulin (1/3000; Molecular Probes, 701265), mouse anti-glucagon (1/1000; Sigma, G2654), rabbit anti-glucagon (1/200; DAKO, A0565), mouse anti-somatostatin (1/200; BCBC Ab1985), rabbit anti-somatostatin (1/200; DAKO, A0566), goat anti-somatostatin (1/200; Santa Cruz Biotechnology, sc-55565), rabbit anti-GFP (1/400; Molecular Probes, A11122), chicken anti-GFP (1/500; Abcam, ab13970), mouse anti-Ppy (1/200; Y. Fujitani[70], mouse anti-Ppy (1/1000; R&D Biosystems, MAB62971), mouse anti-Pyy (1/1000; Abcam, ab112474), rabbit anti-Chga (1/200; Abcam, ab68271) and rabbit anti-Iapp (1/500; Abcam, ab254259). Secondary antibodies were coupled to Alexa 488, 405, 568, 647 (1/500; Molecular Probes) or TRITC, FITC, Cy3 and Cy5 (1/500; Southern Biotech). All antibodies are listed in Supplementary Table 2. Sections were also stained with DAPI. All sections were examined with a confocal microscope (Leica TCS SPE).

**Statistics and reproducibility**. Error bars represent s.d. or s.e.m as indicated in the figure legends. Statistical analyses were performed using Prism v8.0 software applying Mann–Whitney two-sided tests for comparison. *P* values are described in figure Legends. Only the significant statistical differences are indicated in the figures. More than three mice per condition and experiment were analyzed as indicated in the figure legends and Supplementary Tables. Immunofluorescence was performed more than once for each mouse with >4 cryo-sections/mouse. The RNA-seq experiment/reaction was performed once (Fig. 4, Supplementary Fig. 9). The mouse single-cell RNA-seq experiment/reaction was performed twice (Figs. 4 and 5; Supplementary Figs. 7 and 9). The human single-cell RNA-seq experiment/reaction was performed three times based on tissue availability (Fig. 6, Supplementary Fig. 10). Quantitative PCRs were performed twice, using 3-8 individual biological samples as indicated in figure legends; each biological sample was run in triplicate (Figs. 2 and 7). Immunofluorescence for different antibodies was performed once for each mouse with more than 6 cryo-sections/animal being stained at once and analyzed (Figs. 1f, 2d, 6a, 7g, and 8g; Supplementary Figs. 1a, 2, 4a, f, 8, and 11c). The immunofluorescence reaction was repeated more ≥2 for Figs. 1d, 3a, 8d and Supplementary Figs. 3a, 5a, b, 12b.

**Reporting summary**. Further information on research design is available in the Nature Research Reporting Summary linked to this article.

## Data availability
The mouse scRNA-seq and the gamma- and delta-cell bulk RNA-seq dataset generated in this study have been deposited in the NCBI GEO database with the accession number GSE156665. The mouse bulk-RNA-seq of alpha- and beta-cells was obtained from the NCBI GEO database (accession number GSE155519). The human scRNA-seq data are available under restricted access since it is not yet published, access can be obtained upon publication/request in the NCBI GEO database (accession number GSE150724).

Source data for Figs. 1–8 and Supplementary Information are provided with this paper in a Source Data file, availability of associated source data is indicated in each figure legend. Donor details for samples obtained through nPOD are available from the corresponding author upon reasonable request due to donor privacy.

All data and materials used are available from the authors or from commercially available sources. These data are available from corresponding author on reasonable request. Source data are provided with this paper.

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

## Acknowledgements

We thank G. Gallardo and C. Gysler for technical help, S. Chera for the scientific discussions, J.-P. Aubry-Lachainaye for FACS assistance and M. Docquier for RNA-sequencing advices and support. This work was funded with grants from the Swiss National Science Foundation (#310030_192496), the Fondation Aclon and the European Research Council ("Merlin", #884449) to P.L.H. This research was performed with the support of the Network for Pancreatic Organ donors with Diabetes (nPOD; RRID: SCR_014641), a collaborative type 1 diabetes research project supported by JDRF (nPOD: 5-SRA-2018-557-Q-R) and The Leona M. & Harry B. Helmsley Charitable Trust (Grant # 2018PG-T1D053, G-2108-04793). The content and views expressed are the responsibility of the authors and do not necessarily reflect the official view of nPOD. Organ Procurement Organizations (OPO) partnering with nPOD to provide research resources are listed at http://www.jdrfnpod.org/for-partners/npod-partners/."

## Author contributions

M.P.F. conceived and performed the experiments and analyses, and wrote the manuscript; L.v.G analyzed all transcriptomic data, and edited the manuscript; M.V.A. performed and analyzed the gamma-cell ablation experiments; V.C. performed the alpha, beta and delta-cell bulk RNA-seq experiments; K.F. conceived the experiments, edited the manuscript, and generated the human islet scRNA-seq dataset; E.B.T contributed to the generation of the human islet scRNA-seq dataset, and edited the manuscript; D.O. contributed to discussions and wrote the manuscript; T.C contributed to the generation of the Ppy-rtTA mouse line; Y.F generated the Ppy antibody; F.T and P.L.H generated the Ppy-rtTA mouse line, conceived the experiments, interpreted the observations and wrote the manuscript.

## Competing interests

The authors declare no competing interests.
