## [Peer Review File · Nature Communications]

Reviewers' Comments:

Reviewer #1:

Remarks to the Author:

In this manuscript, Perez-Frances et al. generated new mouse lines to explore the characteristics of pancreatic PPY-expressing γ -cells. Their study suggests that the adult γ -cells in mice are derived from embryonic Ppy-expressing cells, and a significant proportion of Ppy+ cells are bihormonal at the mRNA and protein levels. Then, the authors applied scRNA-seq (10x Genomics) to characterize the transcriptomic features of mono- and bihormonal Ppy-expressing cells. In addition, the authors attempted to examine the function of γ -cells, however, the inactivation of Ppy gene and conditional ablation of γ -cells have no effect on glycemia or body weight. Interestingly, the authors finally showed that Ppy-expressing cells engage insulin production after β -cell loss.

Although the authors described the transcriptomic profiles of γ -cells and bihormonal cells using scRNA-seq, the findings throughout the paper are not significant in novelty, and the conclusion that " γ -cells exhibit a functional cell plasticity" is not convincing.

Major concerns:

1. A large number of studies including single-cell transcriptomic analysis have shown that bihormonal endocrine cells exist in humans and mice, and have identified the endocrine lineage ID genes (DOI: 10.1016/j.cell.2020.03.062, 10.1016/j.cels.2016.08.011, 10.1007/s004180050401, 10.1007/s00125-009-1570-x, 10.15252/embr.201540946, 10.1016/j.cels.2016.09.002, 10.1016/j.cmet.2017.04.003, 10.1242/dev.165480, 10.4161/19382014.2014.982949, 10.1038/s41420-017-0014-5, 10.1242/dev.173716, 10.1073/pnas.1602306113, etc.).
2. In Fig.2C, the percentages of adult hormone+ cells labelled with YFP are very high. Did the authors assess the leakage of the Ppy-iYFP system?
3. Line 161-162, the authors demonstrated a significant proportion of the adult γ -cell population is bihormonal, which remains constant even in aged mice. The statistics of each bihormonal Ppy-expressing population from the aged mice should be shown.
4. In supple. Fig 2, the authors showed different distribution of γ -cells in the dorsal and ventral pancreas. However, there is no introduction of the dorsal and ventral pancreas in the text. The authors did not specify which part of the pancreas the used islets came from. Do cells from different parts of the pancreas have transcriptome heterogeneity?
- (5) The authors classified bihormonal cells mainly based on the expression patterns of hormone genes. However, a large portion of Ppy-expressing bihormonal cells are clustered with the monohormonal cells (Fig 4c). This apparently inaccurate cell classification inevitably interferes with subsequent analysis of DEGs.
- (6) DEGs of each islet cell type and transcriptomic signatures of bihormonal cells should be verified by immunostaining or other methods.
- (7) Why are the proportions of human PPY+GCG+/PPY+SST+/PPY+INS+ cell not consistent between immunofluorescence and single-cell RNA-seq data (Fig. 5b, c)? Did authors perform doublet removal in human scRNA-seq data?
- (8) The authors detected insulin expression in YFP-labeled cells after β -cell loss. Are these cells derived from monohormonal Ppy-expressing cells or Ppy+Gcg+ or Ppy+Sst+ cells? How the converted cells are similar to normal β -cells? To make the conclusion convincing, the authors should define a cell fate transition trajectory using scRNA-seq datasets.

Minor concerns:

1. For lineage tracing by Ppy-YFPi, how long was the labelling efficiency was detected after Dox treatment (Line 105-108)?

2. A method for "human islets sorting by novel antibody-based protocol" (line 221) should be described.
3. The distribution of YFP+ and YFP- cells should be presented on the UMAP plot in Fig 4.
4. The authors should show UMAP result of human scRNA-seq data.

Reviewer #2:
None

Reviewer #3:
Remarks to the Author:

Perez-Frances and colleagues present extensive data on the characterization of the "elusive" PPY cell in mouse and human islets. Using CRISPR technology they generated PPY-rtTA knock in mice that can be used for inducible lineage tracing and inactivation. They combine immunostaining, scRNA seq and its analysis for characterization of the PPY cells. They present data on the embryonic labelling with the conclusion that there are bihormonal cells and that there are no newly formed PPY+ cells after birth. After mating with reporter mice and labeling at 1 month of age, they sort the YFP cells for scRNA, with the finding that there are a significant proportion of the labelled cells are bihormonal and an extensive comparison of the monohormonal and bihormonal cells are given. Examining 2 NPOD samples of non-diabetic human pancreas, they similarly find bihormonal PPY + cells but mainly with glucagon and even fewer than in the mouse. In experiments similar to what they have shown previously with massive ablation of the beta cells using Diphtheria toxin in RIP DTR transgenic mice, they find some beta cells that had previously been labelled as PPY + cells, showing again some plasticity in the adult islet cells

The PPY cell has been neglected and so there is value in their characterization. While there are extensive data presented, much are given in a non-user friendly way. For example, to have any indication of the number of cells for any data panel, one must search the long file of source data for Fig 1. The data on mice after the ablation of the PPY cells are only given in the supplemental data, which is inappropriate. Unless the data are given in the manuscript, the results should not be in the abstract, results and discussion. Overall extensive rewriting is needed. Some of the other issues include:

1. In the mouse there are 6 genes identified as PPY ID genes in addition to the introduced YFP and rtTA, but there is no analysis of what are their characteristic transcription factors? PPY is said to be the only ID gene in common with mouse and human; really?
2. As well known in the field, most PPY antibodies cross react with others of the PP fold family. It would be important to show that both antibodies used (one not with published reference; the other commercial) do not cross react with PYY. This is easily done by staining colon sections in which PYY is expressed but PPY is not.
3. In ST1g referring to Fig 4 and the occurrence of bihormonal cells, why do the number of bihormonal + monohormonal cells not reach the total number of cells counted for that hormone. The differences range from 5% for PPY + to 14% for glucagon +.
4. In the ablation experiment after the 6 weeks of regeneration, 37% of all INS+ cells are YFP and 209 Ins+ cells from 8 mice, which calculates a total of 525 Ins+ cells for the 8 mice or 66 Ins+ cell per mouse. What were the glycemic levels of these mice through the regeneration period? Is it possible that the severe hyperglycemia lead to leakiness of the transgene expression? There was no source data given on these important experiments.
5. p 6 "no evidence of postnatal PPY+ cell neogenesis". Wouldn't you then expect a decrease in the percentage of labelled cells since there is massive growth of the pancreas between birth and 9 months of age?

6. There are a degree of sloppiness in the text. The Supplemental tables are often mislabelled eg, p14 data on human donors is said to be ST1n but is actually ST1o; there is no Supplemental Table 5 given as cited as source data for Fig 5. There is no information about the human scRNA data used for comparison from a publication in review. p 4106/p5107 unclear what is meant by "no YFP+ cell was PPY +"; these might refer to the homozygous transgenic mice but that is not stated as such. While it is stated that the human PPY cells have 24 ID genes, these are not given, which makes it difficult to evaluate that the only PPY ID gene in common with the mouse is PPY. Many of the supplemental tables are hard to understand what is being compared. The last column of Suppl Fig 8H-J should be % PYY+/YFP+ cells. It is inappropriate to use 4 significant figures for % of labelled cells and for fold change in qPCR data, In ST1s, with Dox 3.6% of the YFP+ cells are Insulin + but without treatment 2.8% are. Presumably the statistics are based on the number of animals and not the number of cells counted.

REVIEWER COMMENTS

Reviewer #1 (Remarks to the Author):

In this manuscript, Perez-Frances et al. generated new mouse lines to explore the characteristics of pancreatic PPY-expressing γ -cells. Their study suggests that the adult γ -cells in mice are derived from embryonic Ppy-expressing cells, and a significant proportion of Ppy+ cells are bihormonal at the mRNA and protein levels. Then, the authors applied scRNA-seq (10x Genomics) to characterize the transcriptomic features of mono- and bihormonal Ppy-expressing cells. In addition, the authors attempted to examine the function of γ -cells, however, the inactivation of Ppy gene and conditional ablation of γ -cells have no effect on glycemia or body weight. Interestingly, the authors finally showed that Ppy-expressing cells engage insulin production after β -cell loss.

Although the authors described the transcriptomic profiles of γ -cells and bihormonal cells using scRNA-seq, the findings throughout the paper are not significant in novelty, and the conclusion that " γ -cells exhibit a functional cell plasticity" is not convincing.

The description of the transcriptomic profile of Ppy-expressing cells is not the only novelty of this study, but also the following findings:

- The demonstration of the embryonic origin of Ppy-expressing cells, revealing that there is no postnatal γ -cell neogenesis.
- The assessment of i) Ppy gene role and ii) γ -cell requirement in maintaining blood glucose homeostasis and β -cell function, using *in vivo* and *in vitro* functional assays.
- The demonstration that adult γ -cells spontaneously engage insulin expression in situations of β -cell loss and insulin deficit, using inducible cell tracing techniques.

Overall, this study provides new fundamental knowledge relevant to islet cell biology and diabetes, and may have direct implication for the development of future therapies (see the new discussion).

Concerning the lack of novelty and not being convincing, reviewer #2 disagrees, for s/he writes: "In this manuscript, a new knock-in mouse line, YFPI transgenic lines allows efficient, inducible and irreversible labeling of PP-expressing cells and inactivation of PP in a homozygous state. In all subsequent experiments, at characterization of the PP-expressing γ -cells, the PP-rtTA allele was maintained at the heterozygous state to preserve mono allelic wild-type PP expression in PP-YFPI mice. This animal model and subsequent experimental strategy provides good step by step evidence to justify the main concept of this study."

Reviewer #3 emphasizes: "The PPY cell has been neglected and so there is value in their characterization."

Major concerns:

(1) A large number of studies including single-cell transcriptomic analysis have shown that bihormonal endocrine cells exist in humans and mice, and have identified the endocrine lineage ID genes (DOI: 10.1016/j.cell.2020.03.062, 10.1016/j.cels.2016.08.011, 10.1007/s004180050401, 10.1007/s00125-009-1570-x, 10.15252/embr.201540946, 10.1016/j.cels.2016.09.002, 10.1016/j.cmet.2017.04.003, 10.1242/dev.165480, 10.4161/19382014.2014.982949, 10.1038/s41420-017-0014-5, 10.1242/dev.173716, 10.1073/pnas.1602306113, etc.).

Bihormonal endocrine cells have been reported by us and others, at mRNA and protein levels, in embryonic and adult murine and human islets. Yet in the previous transcriptomic analyses cited by the reviewer, these bihormonal cells were either not characterized or discarded due to their low abundance, ambiguous profiles or the high probability of being doublets (see table below).

Here, we have characterized in depth for the first time the identity of the monohormonal and bihormonal pancreatic γ -cells, in mice and humans:

- We have defined both monohormonal Ppy⁺ cells based on a list of identity markers, which include transcription factors and cell surface markers.
- We have also identified a list of functional signaling pathways that are regulated (activated/inhibited) similarly in both mouse and human Ppy-expressing cells.
- We unambiguously identified bihormonal Ppy⁺ cells in mice (at both protein and mRNA levels) and characterized their cell identity. These cells have a hybrid transcriptomic profile (mix of 2 monohormonal cell transcriptomes).
- Similarly, we showed that human islets contain at both protein and transcript level bihormonal Ppy-expressing cells and that those do also have a mix transcriptomic landscape.

Ref. cited by Reviewer #1	Species	Age	Type of study	Bihormonal cells	Limitations of the studies	Novelty of our study
10.1016/j.cell.2020.03.062	Mouse	Adult	scRNA-seq	Ins2+Cck+ ; Ins2+Gcg+ ; Ins2+Sst+	1. None identified bihormonal Ppy+ cells at both transcript and protein level. 2. Bihormonal cells were frequently excluded from the analysis due to low abundance. 3. Any study performed a characterization of the identity of the bihormonal cells. 4. None of the studies compared the transcriptomic profile of monohormonal and bihormonal cells 5. There is no quantification of the proportion of bihormonal cells	1. We have identified Ppy+ bihormonal cells in both species: mouse and human 2. We identified the bihormonal cells at both protein (IF) and transcript level (scRNA-seq) 3. We validated the presence of bihormonal cells in a non-transgenic (WT) mouse 4. We quantified the abundances (proportion) of bihormonal cells in both species. 5. We characterized and compared the identity of the Ppy bihormonal and monohormonal cells 6. We have defined the transcription factors and cellular markers that define the Ppy population 7. We identified the signaling pathways that are activated/inhibited in Ppy cell population
10.1016/j.cels.2016.08.011	Human and mouse	Adult	scRNA-seq	Do not mention		
10.1007/s004180050401	Human	Fetal	IHC	Multiple polyhormonal cells		
10.1007/s00125-009-1570-x	Murine β -cells	Adult, neonatal and fetal	scRNA-seq	Multiple polyhormonal cells		
10.15252/embr.201540946	Human	Adult	scRNA-seq	Ins+Gcg+		
10.1016/j.cels.2016.09.002	Human	Adult	scRNA-seq	Do not mention		
10.1016/j.cmet.2017.04.003	Murine α and β -cells	Adult, neonatal and fetal	scRNA-seq	Ppy+Gcg+ , Ins+Gcg+ , Ins+Sst+ , Ins+Ppy+ , Gcg+Sst+ , Gcg+Ghr+		
10.1242/dev.165480	Human	Fetal	scRNA-seq	Multiple polyhormonal cells.		
			IF	Ins+Gcg+ , Ins+Sst+ , Ghr+Gcg+		
10.4161/19382014.2014.982949	Human	Fetal	IHC	Ins+Gcg+ , Ins+Sst+ ; Ins+Gcg+Sst+ ; Gcg+Ppy+		
10.1038/s41420-017-0014-5	Human	Adult	scRNA-seq	Ins+Gcg+ , Ins+Sst+ , Ins+Ppy+ , Ins+Sst+Ppy+		
			IF	C-peptide+Gcg+, C-peptide+Sst+		
10.1242/dev.173716	Mouse	Fetal	scRNA-seq	Gcg+Ins2+ , Ins2+Ppy+ and Ins+Sst+ were the most abundant		
10.1073/pnas.1602306113	Mouse	Adult	scRNA-seq	Gcg+Ppy+ , Gcg+Ins2+ , multiple other bihormonal and polyhormonal cells		
			RNA FISH	Ppy+Gcg+		

(2) In Fig.2C, the percentages of adult hormone+ cells labelled with YFP are very high. Did the authors assess the leakage of the Ppy-iYFP system?

Using the new Ppy-YFPi mouse model, we can efficiently label embryonic cells that engage Ppy expression during a given gestational DOX pulse period (Suppl. Fig. 4). Therefore, every single cell that activates the *Ppy* gene promoter in presence of DOX during development (pulse period from E7.5 to

E19.5) becomes irreversibly YFP-labelled (Fig. 2C). In fact, YFP-labelling in this model is genetic, and therefore maintained even if the labeled cells stop expressing Ppy (Fig. 2D, and the figure below). Back to the last century (27 years ago!), we published that the ablation of embryonic Ppy-expressing cells leads to a significant reduction in the number of Gcg-, Ins- and Sst-expressing cells (DOI: 10.1073/pnas.91.26.12999). This is in line with results reported now in Fig.2C, demonstrating that a fraction of the α -, β - and δ -cell populations in adults derive from cells having expressed Ppy in the fetal pancreas.

Of note, in Suppl. Fig. 1 we show that there is no leakage of the Ppy-YFPi system by quantifying the percentage of Ppy⁺ cells co-expressing YFP in absence of DOX treatment (No DOX administration, two-months old mice: n=4, 111 islet section, 0.11±0.02 YFP+/islet section. No DOX administration, one-year old mice: n=4, 143 islet sections, 0.1±0.02 YFP+/islet section).

Figure not shown. Presence of YFP-traced hormone⁺ cells that do not maintain Ppy expression 30 days after birth. Immunofluorescence of YFP-traced cells (green) co-expressing Glucagon (red, left panel), Somatostatin (red, middle panel) or Insulin (red, right panel) having lost Ppy expression (cyan). Scale bar: 20 μ m (10 μ m in insets).

(3) Line 161-162, the authors demonstrated a significant proportion of the adult γ -cell population is bihormonal, which remains constant even in aged mice. The statistics of each bihormonal Ppy-expressing population from the aged mice should be shown.

There are no significant differences in the percentages of bihormonal Ppy-expressing cells in one-month-old and one-year-old mice. Statistics were added to Figure 3B (see graph and legend).

(4) In supple. Fig 2, the authors showed different distribution of γ -cells in the dorsal and ventral pancreas. However, there is no introduction of the dorsal and ventral pancreas in the text. The authors did not specify which part of the pancreas the used islets came from. Do cells from different parts of the pancreas have transcriptome heterogeneity?

We analyzed both ventral and dorsal pancreas and observed bihormonal Ppy-expressing cells in the two regions, indicating that γ -cells are heterogeneous independently of the pancreatic primordium of embryonic origin (see Figure below). Similar to the ventral pancreas, Gcg⁺Ppy⁺ and Ins⁺Ppy⁺ cells were

the most and the least prevalent bihormonal cells, respectively, in the dorsal pancreas (YFP⁺Gcg⁺, n=299 cells; YFP⁺Sst⁺, n=94 cells; YFP⁺Ins⁺, n=40 cells) (see Figure below). Bihormonal cells are more frequent in the dorsal compared to ventral pancreas (see quantification in panel b). Of note, the extremely low abundance of Ppy⁺ cells in the dorsal pancreas makes the quantification of the bihormonal population rather challenging. Overall, except for their abundance, no significant differences were observed between ventral or dorsal γ -cells. Thus, we focused on the head/ventral region (mentioned in material and methods), where Ppy⁺ cells are more prevalent. For clarity, we have incorporated in each figure legend of the manuscript the region of the pancreas analyzed. The single cell RNA-sequencing of both mice and human islets (Fig. 4-6) was performed using whole pancreata.

Figure not shown in the paper. γ -cell heterogeneity in the dorsal pancreas of Ppy-YFPi mice.

Immunofluorescence on dorsal pancreatic sections from 2-month-old Ppy-YFPi mice stained with YFP (green) in combination with: Glucagon (red, top left), Somatostatin (red, top right) or Insulin (red, bottom left). Bihormonal Ppy-Gcg (1), Ppy-Sst (2) and Ppy-Ins (3) cells are detected. Scale bars: 20 μ m or 10 μ m (insets 1, 2, 3). **b**, In the dorsal region of the adult Ppy-YFPi mice, 30.1%, 20.1% and 4.6% of YFP-traced cells are Gcg⁺, Sst⁺ and Ins⁺, respectively. For reference, data from ventral pancreas was obtained from Fig 3B (one-month-old mice) of the manuscript. Error bars denote s.e.m.; n=6 mice. Two-tailed Mann-Whitney test (ns p>0.05; * p<0.05; ** p<0.01).

Dorsal Pancreas (Ppy-YFPi) (n=6)						
Scored YFP cells	Scored YFP+Gcg+	Gcg+ / YFP+ cells (% \pm SEM)	Scored YFP+Sst+	Sst+ / YFP+ cells (% \pm SEM)	Scored YFP+Ins+	Ins+ / YFP+ cells (% \pm SEM)
1384	299	30.11 \pm 0.032	94	20.06 \pm 0.043	40	4.58 \pm 0.011

Raw data of this figure (not shown in the paper): γ -cell heterogeneity in the dorsal pancreas of Ppy-YFPi mice.

(5) The authors classified bihormonal cells mainly based on the expression patterns of hormone genes. However, a large portion of Ppy-expressing bihormonal cells are clustered with the monohormonal cells (Fig 4c). This apparently inaccurate cell classification inevitably interferes with subsequent analysis of DEGs.

The point is well taken. We have thus applied clustering analysis to define different cell populations (Clusters 1 to 5) and then use hormone expression to identify the endocrine cells in each cluster (Fig. 4b-c). By doing so, we have *in silico* isolated the genuine hybrid bihormonal cells from the bihormonal cells located in monohormonal cell clusters. These latter cells were excluded as they transcriptionally resemble to monohormonal α -, β - and δ -cells, but express Ppy.

We kindly appreciate the comment of the reviewer since the new analysis is now more accurate and robust. Figures 4 and 5 have been entirely revised accordingly.

(6) DEGs of each islet cell type and transcriptomic signatures of bihormonal cells should be verified by immunostaining or other methods.

In this manuscript, we use single cell RNA-seq to define the transcriptomic signature of bihormonal Ppy-expressing cells by calculating the DEGs between *Ppy*⁺ monohormonal and each of the respective bihormonal populations (105, 14 and 50 DEGs were detected in the *Ppy*⁺*Gcg*⁺, *Ppy*⁺*Sst*⁺ and *Ppy*⁺*Ins2*⁺ cell populations, respectively; Fig. 5). Next, we have defined a list of *Ppy*⁺ bihormonal cell markers (Supplementary Table 1m).

Here, we have verified by immunofluorescence the enrichment of *lapp* and *Chga* in the bihormonal Ppy-producing populations of two independent wild-type (49.6% and 67.9% of *Ppy*⁺*Gcg*⁺ bihormonal cells vs 18.4% and 17.9% of the monohormonal *Ppy*⁺ cells contain *lapp* and *Chga*, respectively; 87.4% and 93.8% of the *Ppy*⁺*Ins*⁺ bihormonal cells vs 8.1% and 28.1% of the monohormonal *Ppy*⁺ contain *lapp* and *Chga*, respectively; minimum islets scored = 22; Suppl. Figure 7 and Supplementary Table 1m). The markers that we validated (*lapp* and *Chga*) were selected based on the availability of commercial antibodies. Of note, the presence of *lapp* and *Chga* in *Ppy*⁺*Sst*⁺ bihormonal cells could not be assessed due to antibody incompatibility (mouse anti-Ppy, rabbit anti-*Chga*, rabbit anti-*lapp* and rabbit or mouse anti-Somatostatin; Supplementary Table 3). Overall, we have confirmed at the protein level the identity of *Ppy*⁺ bihormonal cells in mice.

Incidentally, we observed that both *lapp* and *Chga* staining were stronger in β -cells compared to the bihormonal Ppy-expressing cells. This observation correlates with the single-cell RNA-seq, where both markers are downregulated (in DEGs between *Ppy*⁺*Gcg*⁺ and *Ins2*⁺, *lapp* and *Chga* have a logFC -3.7 and -0.7, respectively; in DEGs between *Ppy*⁺*Ins2*⁺ and *Ins2*⁺, *lapp* has a logFC -0.6, *Chga* was not in the list of DEGs; table below not show).

	Bihormonal Markers	
DEGs	lapp (logFC)	Chga (logFC)
PpyGcg vs Ins2	-3.77	-0.73
PpyIns2 vs Ins2	-0.67	not detected

(7) Why are the proportions of human PPY+GCG+/PPY+SST+/PPY+INS+ cell not consistent between immunofluorescence and single-cell RNA-seq data (Fig. 5b, c)? Did authors perform doublet removal in human scRNA-seq data?

In both human and mouse, a slight increase in the proportion of PPY⁺ bihormonal cells was detected using single-cell RNA-seq as compared to immunofluorescence (mouse: Fig. 3 vs Fig. 5d; human: Fig. 6a-b vs c). This fact is probably due to the higher sensitivity of the transcriptomic techniques compared to the immunofluorescence assays. This is consistent with previous reports, where bihormonal populations were detected at transcriptomic, but not proteomic, level (doi: 10.1073/pnas.1602306113).

We applied two doublet cell removal tools in both mouse and human RNA-seq analysis, DoubletFinder and Scrublet (see methods). By doing so, we accepted the risk of eliminating any actual bihormonal cell

exhibiting a hybrid transcriptional profile. The table below shows the number of detected doublets per endocrine cell population in the human scRNA-seq data (not shown, but we can add it if requested).

Doublets detected						
	Singlet	Only by DFinder	Only by Scrublet	By both	Total doublets	% of doublets
non-horm	2853	131	34	9	174	5,75
GCG	2419	24	9	19	52	2,10
INS	2732	42	31	12	85	3,02
PPY	4058	22	24	24	70	1,70
SST	2352	52	17	9	78	3,21
GCG + INS	13	0	0	1	1	7,14
GCG + PPY	41	3	26	47	76	64,96
GCG + SST	8	0	5	0	5	38,46
GCG + GHRL	2	0	5	2	7	77,78
INS + PPY	21	1	1	0	2	8,70
INS + SST	273	31	45	25	101	27,01
INS + GHRL	0	0	1	0	1	100,00
PPY + SST	28	0	1	0	1	3,45
PPY + GHRL	22	0	26	1	27	55,10
SST + GHRL	0	0	2	0	2	100,00
GCG + PPY + GHRL	7	0	0	5	5	41,67
GCG + INS + SST	1	0	0	0	0	0,00
INS + PPY + SST	2	0	0	0	0	0,00
Total	14832	306	227	154	687	4,43

(8) The authors detected insulin expression in YFP-labeled cells after β -cell loss. Are these cells derived from monohormonal Ppy-expressing cells or Ppy+Gcg+ or Ppy+Sst+ cells? How the converted cells are similar to normal β -cells? To make the conclusion convincing, the authors should define a cell fate transition trajectory using scRNA-seq datasets.

We have demonstrated that adult Ppy-expressing cells engage insulin production after β -cell loss using cell lineage tracing methods (see IF and quantification, Fig 8d-f). To identify whether the new insulin-expressing cells derive from monohormonal Ppy-expressing cells, and/or bihormonal Ppy⁺Gcg⁺ or Ppy⁺Sst⁺ is definitely interesting. Unfortunately, this would require several inexistent mouse models, labeling and tracing specifically each of the Ppy⁺ mono or bihormonal subpopulations. Single-cell RNA-sequencing six weeks after β -cell ablation would provide additional information about the transcriptional profile of the new insulin-producing γ -cells. However, the origin of these cells would remain in doubt, since *in silico* cell fate transition trajectories provide less powerful results than *in vivo* inducible cell lineage tracing.

We never claimed that the converted insulin⁺ cells are similar to β -cells. Instead, we described the engagement of insulin production in YFP-traced γ -cells. We have previously published that human α - and γ -cells have the capability to spontaneously engage insulin expression and display glucose-stimulated insulin secretion responses upon expression of Pdx1 and MafA (doi: 10.1038/s41586-019-0942-8). Similar results were obtained in mouse α - and δ -cells upon β -cell loss (doi: 10.1038/nature08894, doi: 10.1038/nature13633). Interestingly, transcriptomic and proteomic analysis of both murine and human insulin-producing α -cells revealed that a strong α -cell identity was maintained despite the acquisition of β -cell identity features (doi: 10.1038/s41586-019-0942-8, doi: 10.1038/s41556-018-0216-y). Overall, previous data indicate that full α - or γ - to β -cell conversion is not mandatory to achieve glucose-dependent insulin secretion and diabetes relief.

Minor concerns:

1. For lineage tracing by Ppy-YFPi, how long was the labelling efficiency was detected after Dox treatment (Line 105-108)?

Two weeks after DOX treatment, 85% of the Ppy⁺ cells were YFP-traced. Labelling efficiency was unchanged after a chase period of 10 months. This data is included now in Figure 1d-e.

2. A method for “human islets sorting by novel antibody-based protocol” (line 221) should be described.

We have already published this protocol, indeed (doi:10.1038/s41586-019-0942-8; doi:10.1172/jci66514; doi:10.1038/ncomms11756). We have now explained better in the methods section the human islet cell sorting strategy.

3. The distribution of YFP+ and YFP- cells should be presented on the UMAP plot in Fig 4.

We have incorporated in Suppl. Fig. 6 the UMAP distribution of YFP⁺ and YFP⁻ cell fractions. Of note, there is a tight correlation between the distribution of the Ppy⁺ and the YFP⁺ cells.

4. The authors should show UMAP result of human scRNA-seq data.

Here is the UMAP of the human scRNA-seq dataset. This data is part of another manuscript under revision for publication.

Figure not shown. UMAP distribution of the human scRNA-seq dataset. Cells are color-coded based on their identity. Populations of α -, β -, δ - and γ - cells each contain thousands of cells, while the ϵ -cell fraction contains hundreds of cells.

Reviewer #2 (Remarks to the Author):

Reviewer Report for Nature Communication,

The manuscript reports on a study of the phenotype of mouse pancreatic peptide-expressing cell ablation genes, followed by in-depth characterization of these cells. In this study, a well-designed cell-level experiment was carried out to investigate the mixed phenotype, adaptive plasticity and insulin secretion of pancreatic polypeptide (PP) cells. However, in contrast to results from earlier studies, in several animal models lacking the PP gene there were no changes in blood glucose or body weight in the clinical phenotype. In addition, some of these newly designed animal models lack basic animal research detection methods that match the results of cell experiments. These will need to be further improved.

1. In this manuscript, a new knock-in mouse line, YFPi transgenic lines allows efficient, inducible and irreversible labeling of PP-expressing cells and inactivation of PP in a homozygous state. In all subsequent experiments, at characterization of the PP-expressing γ -cells, the PP-rtTA allele was maintained at the heterozygous state to preserve mono allelic wild-type PP expression in PP-YFPi mice. This animal model and subsequent experimental strategy provides good step by step evidence to justify the main concept of this study.

We appreciate that this expert highlights the value of the 3 new mouse models presented (γ -cell tracing, Ppy gene knockout and Ppy⁺ cell ablation) and the subsequent experimental work to characterize and evaluate the functional requirement of Ppy-expressing cells.

2. To my knowledge, this is the first report of inactivation of the PP gene in animal model. However, the phenotype of this animal model appears to differ widely from several PP receptor (NPYY4 receptor knockout) mouse models or the mice of PP receptor pharmacological intervention (for example Y4 agonism) (Gastroenterology,2003, 124: 1325–1336). Y4 agonism with PP in the brainstem is implicated in reducing food intake via indirect effects on gastrointestinal function with the hypothalamus also appearing to be involved (PLoS ONE,2009, 4(12): e8488). Further experiments are required to confirm or explain these differences and hence might have contributed to the phenotype.

There is one published paper describing a constitutive inactivation of Ppy in mice (DOI: 10.1053/j.gastro.2007.08.024). In these KO, food intake and body weight were unaffected, thus nicely correlating with our observations (Fig. 7a-b in the present manuscript). Here, we have evaluated for the first time the impact of Ppy inactivation and γ -cell ablation on blood glucose homeostasis and glucose-stimulated insulin secretion in vivo (Fig. 7 and Suppl. Fig. 10), which was not previously studied.

Alterations in food intake behavior and body weight regulation have only been observed upon genetic or pharmacological inactivation of the Npy receptors, but not upon Ppy gene inactivation (DOI: 10.1053/j.gastro.2007.08.024 and the present study). Of note, the Npy family is a highly redundant family of peptides, which share similar function, protein structure and receptor affinity (PMID: 10949087). Thus, the inactivation of an Npy receptor affects the signaling pathways of all Npy peptides (Npy, Ppy and Pyy) resulting in a more profound phenotype (reduced food intake and body weight). By contrast, Ppy inactivation affects Ppy signaling pathway specifically. This likely explains the “apparent” discrepancies in phenotype between Ppy and Npy receptor KO mice. Of course, we cannot exclude that Ppy inactivation might be compensated, at least in part, by Npy and/or Pyy (γ -cells express Pyy; see Fig. 7).

We have incorporated this relevant observation in the discussion section of the paper.

3. Since generating the adult γ -cells derive from embryonic Ppy-expressing cells is a critical point in this study, it is necessary to describe this experimental procedure in detail, and confirm that YFP labeling activity ceased rapidly after DOX withdrawal and evaluate the residual Cre mRNA expression as readout of DOX clearance in islets of pregnant Ppy-YFPi females. In addition, it is necessary to explain why the chase period was extended up to 9 months after birth in the percentage of YFP-labeled adult cells, but not for other time points? (Fig. 2B, Supplementary Table 1e) And the results need to be provided for there is no evidence of postnatal Ppy+ cell neogenesis (Line 141).

The main focus of our study is to characterize the identity of the adult mouse and human γ -cells. In addition, we took advantage of our model to investigate the origin of this population, thus this is an interesting yet somewhat less critical point of our study.

We have observed that the adult γ -cell population derives from embryonic Ppy-expressing cells, and that there is no evidence of postnatal neogenesis (Fig. 2). Probably, we were not clear enough, and thus we have rephrased the text.

Briefly, we first show that the Ppy-YFPi mouse model is suitable for determining the origin of adult (postnatal) γ -cells. We did this by assessing i) the embryonic Ppy⁺ cell labelling efficiency (Suppl. Fig. 4a-c) and ii) DOX clearance after withdrawal (using Cre expression as a proxy; Suppl. Fig 4d-e). Indeed, embryonic Ppy⁺ cells are efficiently labeled and DOX activity (Cre expression) lasts for just one day after DOX withdrawal (Fig. 2A).

With this system, we show that one month after birth (P30), the fraction of labelled Ppy-expressing cells is the same (approx. 85%) using two different labelling pulses: DOX administration from E7.5 to P30 or DOX from E7.5 to E19.5 (Fig. 2B). This indicates that adult γ -cells originate from Ppy⁺ cells appearing in the embryonic pancreas (i.e. before birth). To evaluate Ppy⁺ cell neogenesis after P30, we also extended the chase period up to 9 months. Here again, the percentage of labeled cells remains unchanged in the short and long chase periods (Fig. 2B: P30 vs 9 months), further suggesting that there is no postnatal neogenesis.

4. The authors should clarify their conclusion “Ppy gene inactivation does not affect blood glucose levels or body weight (Line 248). Previous studies have showed that PP-initiated signaling through NPY6 receptor in VIP neurons regulates the growth hormone axis and body composition (Cell Metabolism, 2014, 19, 58–72). In this study, Npy6r knockout mice demonstrated a significant reduction in lean body mass, suggesting that glucose metabolism might be impaired in these mice. Furthermore, after 12 weeks of high fat diet (HFD), Npy6r knockout mice displayed markedly higher blood glucose and serum insulin levels in response to i.p. glucose injection, signifying impaired glucose metabolism. Together, this data suggests that deficiency in the lack of PP signing in mice exacerbates diet-induced obesity and promotes the associated abnormalities in glucose homeostasis. Therefore, it is necessary to add one more experiment involving high fat diet induced models to confirm this important issue.

As explained above (see answer to question #2), the phenotype of an Npy receptor knockout/overexpression does not necessary correlate with that of the Ppy hormone inactivation, due to ligand/receptor redundancy issues.

The γ_6 receptor ligands are not well defined. In 1996, Weinberg et al. claimed that the major agonists of the γ_6 receptor were Npy and Ppy, being the γ_6 an homolog of the Npy1r isotype (doi: 10.1074/jbc.271.28.16435). Shortly thereafter, Gregor et al. described a different pharmacological affinity for the γ_6 receptor, being more affine to PPY than to the other NPY family members (doi:

10.1074/jbc.271.44.27776). A more recent study revealed that the pharmacology of this receptor is completely distinct from the other known NPY receptors, and suggested that it might not be involved in appetite regulation (doi: 10.1016/s0014-2999(00)00255-7). Therefore, we cannot extrapolate whether Ppy is responsible of the phenotype observed in the cited article (doi: 10.1016/j.cmet.2013.11.019) or whether it could be due to Npy and/or Ppy.

This referee is asking for an HFD experiment on Ppy KO mice. This experiment has been already conducted with the Ppy KO previously reported (see above the answer to point 2; DOI: 10.1053/j.gastro.2007.08.024). In this study, they showed that male and female Ppy-deficient mice treated with high-fat diet display no change in body weight gain or body composition compared with wild-type littermates (mentioned as “data not shown”). This observation is in line with the absence of phenotype in mice in which the hormone or the γ -cells were inactivated and ablated, respectively (Fig. 7 and Suppl. Fig. 10). We have improved the discussion section in the manuscript to emphasize the scientific context of our results.

5. (In the result section 251-261), authors presented: “Because.....About 90% of the YFP+ population co-expressed Ppy in Ppy-rtTA HTZ mice. The expression of Ppy was not impacted upon Ppy gene KO. our results suggest that Ppy absence is well tolerated and has no impact on body weight or blood glucose regulation, at least under basal conditions. It seems this conclusion is a misinterpretation since peptide YY3-36 (PYY3-36), a Y2R agonist, is released from the gastrointestinal tract postprandially in proportion to the calorie content of a meal. It has been reported that peripheral injection of PYY3-36 in rats inhibits food intake and reduces weight gain (Nature2002,418(6898):650-654.)

Indeed, Ppy₃₋₃₆ is released by gut L-cells after food intake. Here, we have observed that ~90% of the YFP-labelled γ -cells express Ppy₃₋₃₆ in both Ppy-YFPi and Ppy-knockout mice (Fig. 7g-h). So, we cannot exclude that intra-islet secreted Ppy₃₋₃₆ could compensate the lack of Ppy action in Ppy KO islets. Yet, γ -cell ablation did not affect glucose homeostasis and body weight regulation (Suppl. Fig. 10d-i). Hence, if a compensatory mechanism takes place in these β -cell ablated mice, it should derive from intestinal Ppy₃₋₃₆ expressing L-cells.

6. Insulin tolerance test is presented as a test to determine the sensitivity of insulin-responsive tissues. In this study, a key finding is that two animal models have the adaptive plasticity to engage insulin production. In order to confirm this result, it would be better to perform one more important experiment (Insulin tolerance test) for the two groups of mice.

β -cell ablation in RIP-DTR mice results in severe hyperglycemia (>35mmol/l) (see also the answer to reviewer #3, question #4). For this reason, insulin therapy (we use subcutaneous insulin implants), is mandatory to keep these animals alive, as reported (doi: 10.1038/nature08894). Consequently, the uncontrolled release of insulin by the implants in diabetic mice would compromise the interpretation regarding the insulin sensitivity after injecting insulin for ITT. This interesting experiment is not feasible in our experimental conditions, as could lead to deadly hypoglycemia.

Reviewer #3 (Remarks to the Author):

Perez-Frances and colleagues present extensive data on the characterization of the "elusive" PPY cell in mouse and human islets. Using CRISPR technology they generated PPY-rtTA knock in mice that can be used for inducible lineage tracing and inactivation. They combine immunostaining, scRNA seq and its analysis for characterization of the PPY cells. They present data on the embryonic labelling with the conclusion that there are bihormonal cells and that there are no newly formed PPY+ cells after birth. After mating with reporter mice and labeling at 1 month of age, they sort the YFP cells for scRNA, with the finding that there are a significant proportion of the labelled cells are bihormonal and an extensive comparison of the monohormonal and bihormonal cells are given. Examining 2 NPOD samples of non-diabetic human pancreas, they similarly find bihormonal PPY + cells but mainly with glucagon and even fewer than in the mouse. In experiments similar to what they have shown previously with massive ablation of the beta cells using Diphtheria toxin in RIP DTR transgenic mice, they find some beta cells that had previously been labelled as PPY + cells, showing again some plasticity in the adult islet cells. The PPY cell has been neglected and so there is value in their characterization. While there are extensive data presented, much are given in a non-user friendly way. For example, to have any indication of the number of cells for any data panel, one must search the long file of source data for Fig 1. The data on mice after the ablation of the PPY cells are only given in the supplemental data, which is inappropriate. Unless the data are given in the manuscript, the results should not be in the abstract, results and discussion. Overall extensive rewriting is needed.

Because Ppy-expressing cells have been neglected so far, we decided to perform their systematic and comprehensive characterization. We bring here a huge amount of data, covering many aspects of γ -cell biology, namely: i) embryonic origin, ii) identity characterization by immunofluorescence and bulk and single-cell RNA-seq, iii) functional characterization of Ppy (hormone) and γ -cells, and iv) plasticity potential after β -cell loss. Unfortunately, because of size constrains, we could not include all the data as main figures. But in this revised version we have reconsidered two of the supplemental figures (old Suppl. Fig. 7 and old Suppl. Fig. 10) and implemented them as main (new Figs. 4 and 7). We have therefore reorganized the supplemental and main data for improving clarity, as well as done a significant rewriting of the text.

1. In the mouse there are 6 genes identified as PPY ID genes in addition to the introduced YFP and rtTA, but there is no analysis of what are their characteristic transcription factors? PPY is said to be the only ID gene in common with mouse and human; really?

We have extensively improved the characterization of the γ -cells by implementing a more reliable and stringent methodology.

As requested by referee #1, we have reconducted the transcriptomic analysis of mouse γ -cells. We have introduced two major changes: the monohormonal and bihormonal cell classification based on clustering, and the reanalysis of the γ -cell ID genes based on pair-wise differential expression analyses. In the revised version, we have integrated a bulk RNA-sequencing dataset to have a more extensive characterization of the γ -cell identity profile. By doing so, we could detect 3240 γ -cell ID genes, including 195 transcription factors such as *Arx*, *Ski* and *Fev*. This data has been incorporated in new Figure 4. Of note, all the scRNA-seq analysis was re-processed based on the newly identified islet ID genes.

We have also reassessed the common features between mouse and human γ -cells by overlapping the pair-wise upregulated DEGs in *PPY*⁺ cells vs each other cell type. Briefly, we could detect at least 12 common genes between the two species, including well-known markers and transcription factors such as *PPY*, *TTR* and *ARX*. These results are in the new Supplementary Figure 8a.

2. As well known in the field, most PPY antibodies cross react with others of the PP fold family. It would be important to show that both antibodies used (one not with published reference; the other commercial) do not cross react with PYY. This is easily done by staining colon sections in which PYY is expressed but PPY is not.

This is exactly what was already done for one antibody in Fig 1. In the islets of Ppy-knockout mice we did not detect Ppy (using the mouse anti-Ppy provided by Y. Fujitani; Fig. 1d-f), but Pyy expression was observed (using the mouse anti-Pyy ab112474 from Abcam; Fig. 7g-h). We confirmed these results using a different anti-Ppy antibody (commercial mouse anti-Ppy; new Suppl. Figure 2). Together, this excludes any antibody cross-reaction between both Ppy and Pyy peptides, which represents an important control for the interpretation of the results. Also, because Ppy and Pyy monoclonal antibodies are raised in mouse, we cannot detect both peptides simultaneously.

3. In ST1g referring to Fig 4 and the occurrence of bihormonal cells, why do the number of bihormonal + monohormonal cells not reach the total number of cells counted for that hormone. The differences range from 5% for PPY + to 14% for glucagon +.

The sum of bihormonal and monohormonal cells does not reach the total number of scored cells because multihomonal cells (more than 2 co-expressed hormones) were excluded from the analyses. We decided to exclude these multihomonal cells as they were either present in very low frequency or were not validated by IF. For instance, we failed to detect by immunostaining the most abundant multihormonal cell population in our dataset (*Gcg⁺Ppy⁺Sst⁺*; n=122; a minimum of 20 islet sections were analyzed in ventral and dorsal pancreas, n=2 mice).

To avoid ambiguity, we have modified the table (see new ST1g) adding the multihormonal cells that were excluded in the subsequent analysis.

4. In the ablation experiment after the 6 weeks of regeneration, 37% of all INS⁺ cells are YFP and 209 Ins⁺ cells from 8 mice, which calculates a total of 525 Ins⁺ cells for the 8 mice or 66 Ins⁺ cell per mouse. What were the glycemic levels of these mice through the regeneration period? Is it possible that the severe hyperglycemia lead to leakiness of the transgene expression? There was no source data given on these important experiments.

We have added in Fig. 8c the glycemic levels of β -cell-ablated mice through the regeneration period. Glycemia in diabetic mice fluctuate over time due to the uncontrolled release of insulin from the subcutaneous implants (see answer to point 6 of referee #2).

In this ablation experiment (Fig. 8), hyperglycemia was induced 15 days after DOX withdrawal (15 days after cell labelling), which is enough time for DOX clearance (Suppl. Fig 4e, DOX activity lasts just during one day after DOX withdrawal). Thus, if hyperglycemia leads to transgene leakiness, it must occur in absence of DOX. To assess this possibility, we measured YFP labelling in absence of DOX in hyperglycemic Ppy-YFPi mice. We found no insulin⁺ nor Ppy⁺ cell labelled with YFP in these diabetic mice (see below figure; not shown in the paper; inset shows one bihormonal Ppy⁺Ins⁺ cell that is not labelled with YFP). This clearly indicates that there is no leakiness of the transgenes upon severe hyperglycemia and that YFP⁺Ins⁺ cells represent reprogrammed γ -cells.

Figure not shown. Absence of leakiness in transgene expression upon hyperglycemia. **a**, β -cell ablation triggers hyperglycemia in non-DOX-treated mice. No β -cell ablation: $n=3$. β -cell ablation; no DOX administration: $n=4$. **b**, Most of the Ppy-expressing and the new insulin-expressing cells are traced with YFP upon DOX administration and β -cell ablation. Representative image of control mice (β -cell ablation and DOX administration). Ppy: grey, Insulin: red and YFP: green. **c**, no YFP expression was detected in both Ppy+ and Ins+ cells without DOX administration in β -cell ablated mice (β -cell ablation; no DOX administration: $n=4$). Ppy: grey, Insulin: red and YFP: green. Scale bars: 20 μ m or 10 μ m (insets). Raw data is supplied below.

For your information, the raw data from the regeneration experiments was already added (Suppl. Table 1w). We are adding below the raw data of the experiment assessing the leakiness of the transgene upon hyperglycemia.

Raw data from Figure: No leaky transgene expression upon hyperglycemia.

Condition	Scored Ppy+	Scored Ins+	Scored Ins+GFP+	Scored PPY+GFP+	YFP+/Ppy+ (% \pm SEM)	YFP+/Ins+ (% \pm SEM)
β -cell ablation; No DOX administration	1800	81	0	0	0.00%	0.00%

5. p 6 "no evidence of postnatal PPY+ cell neogenesis". Wouldn't you then expect a decrease in the percentage of labelled cells since there is massive growth of the pancreas between birth and 9 months of age?

A decrease in the percentage of labelled Ppy+ cells (Ppy+YFP+ cells/total Ppy+ cells) would occur only if there is postnatal neogenesis from progenitor cells (appearance of new Ppy+ cells after the time of DOX exposure, for instance after birth). This is not what we observed (Fig. 2), as the percentage of labelled Ppy+ cells remained constant after birth (even after an extended chase period). This strongly suggests that, if the Ppy cell population expands during the growth of pancreas, it relies on the self-duplication of pre-existing Ppy cells labelled during embryogenesis. Of note, exocrine expansion contributes mostly to pancreas growth after birth (endocrine compartment expands to a much less extent; DOI: 10.1016/j.devcel.2018.05.024 and DOI: 10.3109/03009734.2016.1154906).

6. There are a degree of sloppiness in the text.

We would like to apologize for this sloppiness. All mistakes have been corrected in the manuscript.

- **The Supplemental tables are often mislabelled** eg, p14 data on human donors is said to be ST1n but is actually ST1o. *Corrected.*
- **There is no Supplemental Table 5 given as cited as source data for Fig 5.** *Corrected.*
- **There is no information about the human scRNA data used for comparison from a publication in review.** *We provide a temporary token number of the NCBI GEO database (see below). GEO accession number: GSE150724, Token: orirgiaavfwpcn*
- **p 4106/p5107 unclear what is meant by "no YFP+ cell was PPY +"; these might refer to the homozygous transgenic mice but that is not stated as such.** *We have rephrased this sentence in the manuscript.*
- **While it is stated that the human PPY cells have 24 ID genes, these are not given, which makes it difficult to evaluate that the only PPY ID gene in common with the mouse is PPY.** *This data is included in another manuscript also under revision. Below we are confidentially sharing with the reviewer the list of human γ -cell ID genes.*
- **Many of the supplemental tables are hard to understand what is being compared.** *We have edited them to gain clarity.*
- **The last column of Suppl. Fig 8H-J should be % PYY+/YFP+ cells.** *Corrected the figure labelling (see Fig. 7g).*
- **It is inappropriate to use 4 significant figures for % of labelled cells and for fold change in qPCR data.** *We have reduced the number of panels from four to two (Fig. 7).*
- **In ST1s, with Dox 3.6% of the YFP+ cells are Insulin + but without treatment 2.8% are. Presumably, the statistics are based on the number of animals and not the number of cells counted.** *First, we calculate in each mouse the percentage of YFP⁺Ins⁺ cells before or after β -cell ablation (ST1w). Then, we apply statistics to assess significance between the percentages of YFP-labelled ins⁺ cells of the control and β -cell ablated groups (Mann-Whitney tests for comparison). Thus, we base the statistics on the number of cells per each animal. In this particular case (ST1w), we did not detect any significant difference between the two groups ($P=0.662$). We now provide the graphs from Fig. 8e-f in scatter plot to facilitate the visualization of the intra-group variation.*

human γ ID genes
ABCC9
AQP3
BTG2
CALB1
CHRM3
ETV1
FGFR1
FXVD2
FXVD6-FXVD2
GCNT3
ID2
ID4
INPP5F
MEIS2
PAX6
PPY
PTP4A3
PXK
SERTM1
SLC6A4
SLITRK6
STMN2
THSD7A
TPH1

Table. Human ID genes of the pancreatic endocrine γ -cells.

We would like to provide the accession numbers and tokens to the RNA-seq datasets used in this study.

	GEO number	Token
Mouse Single cell RNA-seq dataset	GSE156665	mngpgqisfpundid
Mouse Bulk RNA-seq dataset	GSE156665	mngpgqisfpundid
Human Single cell RNA-seq dataset	GSE150724	origiaavfwvpcn

Reviewers' Comments:

Reviewer #1:

Remarks to the Author:

The authors have addressed my questions. The revised manuscript is appropriate for publication.

Reviewer #2:

Remarks to the Author:

The authors have addressed my concerns and questions satisfactorily. I have no further questions.

Reviewer #3:

Remarks to the Author:

The revised manuscript from Perez-Frances has addressed many of the previously raised criticisms. The have added new Fig 4 and 7, reanalyzed the RNAseq data, added antibody validation, and the blood glucose levels through the ablation experiment. However, some issues still remain.

As previously stated, the quantifications have 4 significant figures which is inappropriate for mean \pm sem for both % cell number and PCR fold change. It seems that the authors did not understand the critique since they said they removed 2 of the 4 panels in Fig 7. To be clear, counting cells at 2 wk to get 84.92 \pm 0.34% should at best be 84.9 \pm 0.3%. Throughout the manuscript, the significant figures need to be corrected to be appropriate.

Figure legends still no not indicate the number of cells counted and often not even the number of mice. Yes, the supplemental data lists all in great detail but the reader should not have to hunt through extensive supplemental data to get an indication of how many cells per mouse were counted and what was the incidence of the different cell types. For example, in the human studies, Suppl Table 1r (Fig 6) shows that by immunostaining in one of the donors there are 2 PPY+GCG+/ 1395 PPY+ cells and for the other 6/1154 with no PPY+INS+ or PPY+SST+ cells and at the transcriptome level there were between 0.6 and 0.8% of the PPY+ cells that were bihormonal (25-37 bihormonal in the 4135 enriched for PPY+ cells from 3 donors). With the actual number of cells indicated, one can understand how rare these cells actually are.

In fact, the numbers should indicate cells/ mouse, not lumped sums for all the mice and mean. Additionally, since most data panels are from less than 10 mice, the data should be given as scatter plots as in Fig 3b with mean indicated rather than histogram.

The new revised Fig4 presents the mouse scRNA seq data. It is unclear why the numbers in panel C does not agree with the source data given in Suppl Table 1g in which the Total number after clustering filter seems to ignore those bihormonal cells in the individual clusters and only use the cluster 5 numbers for PPY-GCG and PPY-SST bihormonal and some unknown number for the PPY-INS2 ones. Cluster 5 is not labeled on the figure but in the text is call the "bihormonal cells". Actually, many of the bihormonal are in the other hormone clusters (1-4) The legend should have some explanation. Additionally, the genes listed in F as uniquely gamma cell ID genes include arx (which is expressed in alpha cells), vegfa (which is expressed in beta cells), epcam (expressed across the islet and pancreas) and then lamp1, a lysosomal enzyme as the top 10 cell surface marker. PYY is also given as a PPY cell identity gene but most reports have PYY expressed in some (20-30%) adult alpha cells and most delta cells. Similarly in Fig 5 why are arx not listed as bihormonal marker for PPY-GCG and Chga for PPY-INS2 cells?

The text (line 329) credits Fig8D, an immunostained image, of showing that all of the PPY+INS + removed by the diphtheria toxin and that at 6 wks after ablation the reprogrammed cells expressed little PPY. These are two important issues are not shown in that one image.

Here characterization of the lack of the PPY cells is done by examining the homozygous mice which would have them eliminated during fetal development. These mice are said not to differ in body

weight or glycemia from WT or the heterozygous. These findings seem to be in contrast to the 1994 ablation of PPY cells in mice that expressed Diphtheria toxin under the PPY promoter in which both the insulin and somatostatin expressing cells were lacking. How are these findings reconciled?

Minor:

Abstract: "PPY+ expressing gamma cells are rarest of pancreatic islet cell type", NO, the epsilon is. "Data" is the plural form, so the verbs should agree.

Fig2 legend states: " The % of YFP labelled PPY cells is taken for reference from Panel E" There is no panel E.

Why are there different number of mice for the 2 tables of ST1w for Fig 8 D-F?

line 173: please specify whether the animals were treated with DOX embryonically, at 1 month or as adults.

line 200: bulk RNAseq for PPY expressing cells but was this actually a comparison of the data from various cell types selected by YFPdriven by each of the hormone drivers as mentioned in the Methods?

REVIEWER COMMENTS

Reviewer #1 (Remarks to the Author):

The authors have addressed my questions. The revised manuscript is appropriate for publication.

Reviewer #2 (Remarks to the Author):

The authors have addressed my concerns and questions satisfactorily. I have no further questions.

Reviewer #3 (Remarks to the Author):

The revised manuscript from Perez-Frances has addressed many of the previously raised criticisms. The have added new Fig 4 and 7, reanalyzed the RNAseq data, added antibody validation, and the blood glucose levels through the ablation experiment. However, some issues still remain.

We are very glad. Reviewers #1 and #2 are satisfied and have no further comments; the remaining issues of reviewer #3 are minor and are fully addressed below. We are grateful to this expert, for the careful analysis and appropriate considerations listed in her/his comments.

As previously stated, the quantifications have 4 significant figures which is inappropriate for mean \pm sem for both % cell number and PCR fold change. It seems that the authors did not understand the critique since they said they removed 2 of the 4 panels in Fig 7. To be clear, counting cells at 2 wk to get 84.92 \pm 0.34% should at best be 84.9 \pm 0.3%. Throughout the manuscript, the significant figures need to be corrected to be appropriate.

We apologize for the misunderstanding. We are now indicating just one decimal throughout the manuscript.

Figure legends still no not indicate the number of cells counted and often not even the number of mice. Yes, the supplemental data lists all in great detail but the reader should not have to hunt through extensive supplemental data to get an indication of how many cells per mouse were counted and what was the incidence of the different cell types. For example, in the human studies, Suppl Table 1r (Fig 6) shows that by immunostaining in one of the donors there are 2 PPY+GCG+/ 1395 PPY+ cells and for the other 6/1154 with no PPY+INS+ or PPY+SST+ cells and at the transcriptome level there were between 0.6 and 0.8% of the PPY+ cells that were bihormonal (25-37 bihormonal in the 4135 enriched for PPY+ cells from 3 donors). With the actual number of cells indicated, one can understand how rare these cells actually are.

We have incorporated in all figure legends the number of mice and cells counted in each experiment.

In fact, the numbers should indicate cells/ mouse, not lumped sums for all the mice and mean. Additionally, since most data panels are from less than 10 mice, the data should be given as scatter plots as in Fig 3b with mean indicated rather than histogram.

We indicate now the number of cells scored per mouse in all supplementary tables.

All graphs are now shown as scatter plots with histograms indicating the mean value of the group.

The new revised Fig4 presents the mouse scRNA seq data. It is unclear why the numbers in panel C does not agree with the source data given in Suppl Table 1g in which the Total number after clustering filter seems to ignore those bihormonal cells in the individual clusters and only use the cluster 5 numbers for PPY-GCG and

PPY-SST bihormonal and some unknown number for the PPY-INS2 ones. Cluster 5 is not labeled on the figure but in the text is call the "bihormonal cells". Actually, many of the bihormonal are in the other hormone clusters (1-4) The legend should have some explanation.

We have corrected the number of *Ppy+Ins2+* cells after clustering filtering in the Suppl Table 1g. We apologize for the mistake.

As indicated in the manuscript (line 224): "By applying clustering analysis, we have in silico isolated the genuine hybrid bihormonal cells (black dots; Fig. 5A-C) from the bihormonal cells located in monohormonal cell clusters (colored dots; Fig. 5A-C). These latter cells were excluded as they transcriptionally resemble to monohormonal α -, β - and δ -cells, but express *Ppy*." Thus, in the Suppl. Table 1g we only included the bihormonal cells located in Cluster 5. We have incorporated in the legend of Fig. 5 why bihormonal *Ppy*-expressing cells in clusters 1-4 were excluded from the analysis.

Additionally, the genes listed in F as uniquely gamma cell ID genes include *arx* (which is expressed in alpha cells), *vegfa* (which is expressed in beta cells), *epcam* (expressed across the islet and pancreas) and then *lamp1*, a lysosomal enzyme as the top 10 cell surface marker. *PYY* is also given as a PPY cell identity gene but most reports have *PYY* expressed in some (20-30%) adult alpha cells and most delta cells.

We define transcriptomic cell identity as the set of genes that define a given islet cell type. These ID genes represent differentially expressed genes between islet cell types and thus are not necessarily specific as you pointed it out. Two cell types or more may expressed the very same gene but at various levels. This gene is considered as an ID gene if it is significantly modulated in a cell type as compared the other islet cells. For instance, *Arx* is expressed in both γ - and α -cells. However, as *Arx* is significantly more expressed in γ -cells as compared to all the others islet cells types, it is considered as a γ -cell ID gene. The same is true for all the genes you mentioned that can be expressed in more than one islet cell type.

To obtain the γ -ID surface markers, we intersected the list of bulk RNA-seq γ -ID genes with a published list of 1296 murine cell surface proteins (DOI: 10.1371/journal.pone.0121314). In fact, *Lamp1* is a lysosome-associated membrane glycoprotein, which has been shown to be present in the cell surface (DOI: 10.1007/s00432-015-1917-2, DOI: 10.1006/cimm.1996.0167). We have incorporated in the Methods section how the γ -ID surface markers were obtained.

Similarly in Fig 5 why are *arx* not listed as bihormonal marker for PPY-GCG and *Chga* for PPY-INS2 cells?

Arx is not differentially expressed in *Ppy+Gcg+* cells compared to monohormonal *Ppy+* cells (Fig. 5f top and in Suppl. Table 1l). These data indicate that *Arx* is similarly expressed in the bihormonal *Ppy+Gcg+* and the monohormonal *Ppy+* populations.

Chga is upregulated in bihormonal *Ppy+Ins2+* cells compared to monohormonal *Ppy+* cells, and it is indicated in Fig. 5f (bottom) and in Suppl. Table 1l.

The text (line 329) credits Fig8D, an immunostained image, of showing that all of the PPY+INS + removed by the diphtheria toxin and that at 6 wks after ablation the reprogrammed cells expressed little PPY. These are two important issues are not shown in that one image.

The strong decrease in YFP-labeled *Ins+* cells 5 days after DT (black vs green dots in Fig. 8E and Fig. 8D; Supplementary Table 1w) indicates that pre-existing bihormonal *Ppy+Ins+* cells were efficiently ablated together with β -cells upon DT. We clarified this point in the text.

Most reprogrammed insulin-expressing cells (95,3%±0.1; n=4; 103 Ins+YFP+Ppy- out of 108 Ins+YFP+ cells scored) do not maintain Ppy after DT (inset in Fig. 8G). The text in the manuscript was modified accordingly for clarity. Panel D of Fig. 8 has been separated in two (new panel G) for clarity.

Here characterization of the lack of the PPY cells is done by examining the homozygous mice which would have them eliminated during fetal development. These mice are said not to differ in body weight or glycemia from WT or the heterozygous. These findings seem to be in contrast to the 1994 ablation of PPY cells in mice that expressed Diphtheria toxin under the PPY promoter in which both the insulin and somatostatin expressing cells were lacking. How are these findings reconciled?

These studies are complementary, and the present work confirms many aspects of the early 1994 study.

In 1994, one of us (Herrera, and co-workers) studied transgenic embryos in which Ppy-expressing embryonic cells in the developing pancreas were constitutively ablated as a consequence of the expression of the diphtheria toxin A (active subunit of the toxin). At late fetal stages, the authors observed that a significant fraction of insulin- and somatostatin-containing cells were lacking, suggesting that their precursors had expressed the Ppy gene during development, or more specifically, that they had activated the Ppy promoter (DOI: 10.1073/pnas.91.26.12999). In that study, no transgenic line was established: these were transient transgenic analyses, where F0 embryos were euthanized and analyzed.

Here, by providing DOX during gestation, embryonic Ppy-expressing cells become irreversibly tagged with YFP, and their progeny can thus be followed until late in life (10 months of age in this study). The fact that a fraction of non-gamma cells is YFP-labeled in adults is therefore compatible with the findings reported in the 1994 paper.

By contrast, in the present study, we eliminate Ppy gene expression in homozygous mice, not the cells in which this gene is expressed. The transgenic lines were generated by a Cas9-mediated knock-in, in the Ppy coding region. In the present study we have generated in parallel a transgenic line to achieve an inducible (conditional) Ppy⁺ cell ablation system, where Ppy-expressing cells bear the diphtheria toxin receptor on their surface (whether they are hemi- or homozygous). Therefore, in this case, Ppy⁺ cells are selectively ablated exclusively upon diphtheria toxin administration, i.e. in adult mice that have otherwise developed normally. As reported in the manuscript, no impact on body weight nor glycemia was observed upon Ppy⁺ cell ablation in adult mice.

Minor:

Abstract: "PPY+ expressing gamma cells are rarest of pancreatic islet cell type", NO, the epsilon is.

We have reworded this sentence: "one of the rarest..." In fact, ghrelin-expressing epsilon cells are not normally found in adult mouse islets, yet they are present in adult human islets.

"Data" is the plural form, so the verbs should agree.

We naturally agree, and have corrected this mistake. In modern use in English, however, it is not treated as a plural and often takes a singular verb.

Fig2 legend states: "The % of YFP labelled PPY cells is taken for reference from Panel E" There is no panel E.

Corrected. Data were taken for reference from panel B.

Why are there different number of mice for the 2 tables of ST1w for Fig 8 D-F?

The expression of Ppy in the regenerated insulin-expressing cells was assessed in 4 out of the 8 mice (as all 8 mice exhibit very similar phenotype). For clarity, we decided to split the table in two (ST1w).

line 173: please specify whether the animals were treated with DOX embryonically, at 1 month or as adults.

One-month-old Ppy-YFPi mice were treated with DOX for two weeks. The transcriptomic analysis was performed at two-months (2 weeks after DOX withdrawal). This is now mentioned in the text.

line 200: bulk RNAseq for PPY expressing cells but was this actually a comparison of the data from various cell types selected by YFPdriven by each of the hormone drivers as mentioned in the Methods?

Yes, we performed bulk RNA-seq from distinct mouse models in which a specific islet cell type was labelled with a fluorescent reporter (Gcg-YFPi, RIP-CreER, Ppy-YFPi and Sst-YFP for α -, β -, γ - and δ -cells, respectively). We then integrated all these datasets for the present analysis. We have clarified this point in the methods.

Reviewers' Comments:

Reviewer #3:

Remarks to the Author:

Thank you for the completeness of this revision. It is now a very nice contribution to the field.